# ExpoTab: One-Step Mixed-Type Tabular Data Generation using Manifold Learning

## Abstract

Fast generation of high-quality synthetic tabular data is critical for supporting latency-sensitive applications. This includes real-time fraud detection, where rapid model iteration depends on quick data synthesis, and large-scale system testing (e.g., e-commerce load simulation), where generators must produce massive volumes under strict time constraints to assess performance. However, current state-of-the-art (SOTA) tabular generators are inefficient, often requiring numerous denoising steps, and ineffective at handling high-cardinality categorical features and mixed-type data representations common in real-world datasets. We introduce an Exponential map method for Tabular data fast generation(ExpoTab), a geometric-aware flow model that addresses these challenges through two key innovations: (1) the ExpoTab-Encoder, a spherical encoder that projects categorical features onto optimized manifold coordinates, with quantile transformers to unify all features into a bounded representation space for stable optimization; and (2) the ExpoTab-Sampler, presenting a curved-path velocity field that leverages Riemannian geometry to model geodesics as one-step exponential maps, achieving one-step sampling. Extensive experiments across ten benchmarks demonstrate that ExpoTab outperforms existing methods in three key aspects: (a) our manifold-aware encoder achieves superior separability and flow matching performance; (b) the exponential map sampler accelerates generation by 4–10× while maintaining competitive fidelity; and (c) ExpoTab represents the first one-step generator effectively handling mixed-type tabular data, achieving ∼0.1s generation for 1M samples.

## 1 Introduction

The generation of high-quality synthetic tabular data is essential for modern machine learning, particularly in applications demanding scalability and real-time processing. In fraud detection, for instance, a valid system has to pass a test to validate its ability to handle large amount of tabular data within a short time. A fast generation of synthetic data of the same distribution as in real world plays an essential role in the development of modern powerful detection mechanism(Fiore et al., 2019).

While diffusion-based or flow-based methods have achieved state-of-the-art success in generating tabular data (Kotelnikov et al., 2023; Shi et al., 2024; Zhang et al., 2023; Si et al., 2025), their iterative sampling process remains prohibitively slow for these large-scale or latency-sensitive scenarios. The superlinear scaling of sampling time with data volume, as shown in Figure 4, renders existing SOTA generative models inefficient for large-scale or real-

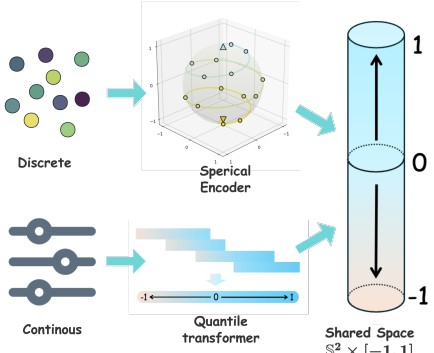

Figure 1: ExpoTab-Encoder for continuous and discrete data encoding

time tabular data generation. This has spurred interest in one-step methods, but the dominant image-based approach—distilling a pre-trained multi-step teacher model—is fundamentally misaligned with tabular needs. This two-stage process is inherently inefficient and introduces a performance

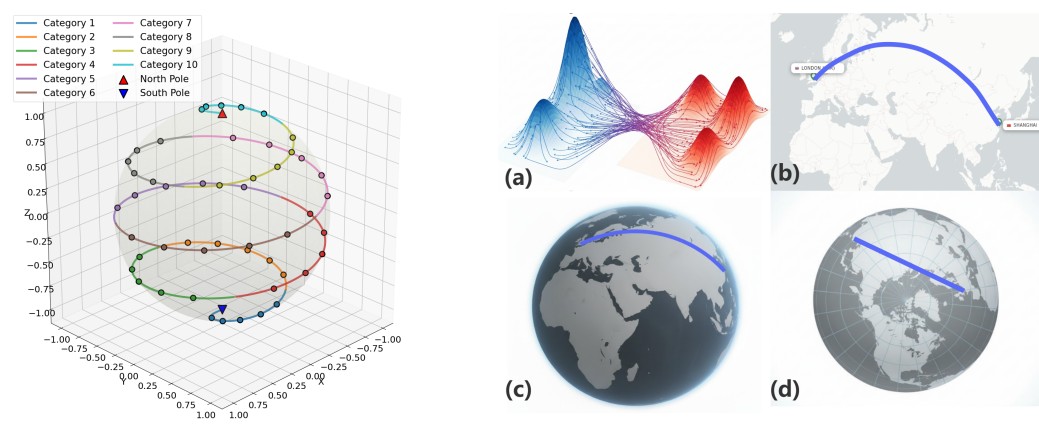

(A) Spherical Encoding    (B) Manifold-aware learning in ExpoTab-Sampler

Figure 2: (A) The discrete catagorical data are maped to a spiral on the unit sphere (B) The ExpoTab-Sampler first uses flow matching to learn a velocity field, creating a path from a source to a target distribution (a). This path is like an airline route between two cities (b). We then learn a curved space where this path becomes a geodesic—the shortest possible route—similar to a flight path on a globe (c). Finally, a learned correction, $f_\theta$, straightens this geodesic into a direct line (d), enabling fast, single-step generation.

ceiling, as the student model only approximates the teacher's output. In contrast, direct one-step methods, such as those learning a mean velocity field (Geng et al., 2025), offer a more direct solution. By being trained end-to-end to regress the direct path from noise to data, they entirely bypass the need for a teacher model and the distillation phase, achieving both superior efficiency and a more robust training objective.

However, the unique characteristics of tabular data pose fundamental challenges for direct application of flow matching methods in one-step generation. Standard flow matching learns velocity fields along straight-line paths from noise to data (Lipman et al., 2023), but this approach struggles with tabular data's heterogeneous feature distributions, which contain both sharp categorical boundaries and smooth continuous distributions. Learning mean velocity along linear trajectories forces the model to approximate complex transitions across diverse data manifolds using simple linear interpolations, ultimately degrading generated data quality. The second is the representation problem: existing approaches struggle to embed mixed categorical and continuous features into a shared space without compromising their distinct characteristics. Sparse representations like one-hot encoding create discontinuous, high-dimensional spaces that are difficult to model effectively (Krishnan et al., 2017; Poslavskaya & Korolev, 2023; Si et al., 2025), while methods like learned embeddings or variational autoencoders often blur categorical boundaries, producing invalid interpolations—such as non-existent hybrid categories (Guo & Berkhahn, 2016; Xu et al., 2019; Zhang et al., 2023; Si et al., 2025). This undermines both the fidelity and utility of the generated tabular data.

To address these challenges, we introduce an **Expo**nential map for **Tab**ular data generation(**ExpoTab**), a novel one-step generative framework specifically designed for mixed-type tabular data. ExpoTab solves the representation problem by providing spherical embeddings for high-cardinality categorical features and combining them with quantile-transformed continuous features in a unified bounded representation space. This combination is crucial because the spherical embedding preserves semantic relationships between discrete categories through geometric proximity on the manifold, while the quantile transformation ensures continuous features are normalized and bounded within [0,1]. Together, they form a structured, compact domain that avoids the sparsity and discontinuity of one-hot encodings and prevents invalid interpolations across feature types. More importantly, ExpoTab solves the generation problem by learning adaptive curved trajectories—geodesics—that naturally conform to the underlying data manifold. These geodesics are particularly well-suited for the unified bounded representation space, as they respect the geometric structure of the spherical embeddings and the boundedness of the continuous representations, ensur-

ing that all generated samples lie within the semantically meaningful domain. By directly approximating the exponential map of the implicit data geometry, ExpoTab achieves one-step sampling that dynamically adjusts the generative path, preserving the fidelity of both categorical and continuous features without traversing off-manifold regions. The principal contributions of this work are shown below:

1. **First One-Step Generator for Mixed-Type Tabular Data:** We introduce the first one-step generation framework for mixed-type tabular data. It achieves rapid generation speeds ($\sim$0.1s for 1M samples) for large-scale datasets containing both discrete and continuous features, enabling real-time applications such as fraud detection and rapid online model validation.

2. **Novel Unified Spherical Encoder (ExpoTab-Encoder):** We propose ExpoTab-Encoder, a unified encoder that uses a spherical representation to enhance the separability of categorical data, particularly for features with high cardinality. This design significantly improves the performance of flow matching on tabular data dominated by discrete features.

3. **Manifold-Based Sampler with Exponential Maps (ExpoTab-Sampler):** We present ExpoTab-Sampler, a one-step generator based on manifold learning that models the generation trajectory as an exponential map. Generation is framed as a constant-velocity motion combined with a learned cumulative correction ($f_\theta$) that accounts for the underlying curved geometry, mimicking the parallel transport of a geodesic. This method accelerates generation by 4x to 10x while maintaining competitive fidelity, utility, and privacy compared to state-of-the-art methods.

## 2 RELATED WORK

### 2.1 TABULAR DATA GENERATION

The field of tabular data generation has evolved from generative adversarial networks (GANs) like CTGAN (Xu et al., 2019) toward diffusion and flow-based models that offer improved training stability and sample quality. Modern diffusion approaches for tabular data, such as TabDDPM (Kotelnikov et al., 2023), typically process numerical and categorical features using separate heads or noise schedules. Several works have introduced specialized mechanisms to handle mixed-type data more effectively: Tabsyn (Zhang et al., 2023) uses a latent diffusion process, while TabDiff (Shi et al., 2024) and CDTD (Mueller et al., 2023) propose adaptive noise schedulers tailored to each feature type. More recently, TabRep-DDPM (Si et al., 2025) and TabRep-Flow unify categorical and continuous representations in a shared continuous space, improving coherence across feature types. While these multi-step denoising methods achieve high-quality generation, their iterative nature results in significant computational overhead, limiting their practical applicability in latency-sensitive scenarios.

### 2.2 REPRESENTATION LEARNING FOR MIXED-TYPE DATA

Another challenge in one-step tabular generation lies in developing effective unified representations for mixed-type data. Existing approaches suffer from several limitations. Early methods used sparse embeddings like one-hot encoding, which create high-dimensional, discontinuous latent spaces that hinder effective interpolation between categories. More recent dense representation methods like Analog Bits (Chen et al., 2022) improve continuity but exhibit poor scalability for high-cardinality features. SOTA embedding approaches like TabRep (Si et al., 2025) have been proposed to unify tabular data in a continuous space by mapping categorical features onto a circle. However, these circular embeddings face significant limitations: they provide limited expressiveness for high-cardinality features and demonstrate poor separability under noisy conditions, as demonstrated in Figure 3.

### 2.3 TOWARDS ONE-STEP GENERATION

From a practical standpoint, few-step generation become important for tabular data generation. First, edge deployment scenarios (e.g., medical diagnostics, IoT devices) require low-latency generation. Second, enterprise-scale applications that process millions of records demand sub-linear time com-

plexity to sustain feasible throughput. Recent works compress multi-step diffusion models into one-step generators (Yin et al., 2024; Zhou et al., 2024), but they are computationally expensive due to two-stage distillation. Yin et al. (2024) use *distribution matching*, aligning the student's one-step outputs with the teacher's final distribution, which requires simulating the teacher's full trajectory. Zhou et al. (2024) propose *score identity distillation*, enforcing that the student's score estimates match the teacher's at each step, enabling fast convergence but still needing multi-step teacher evaluations during training.

An alternative direction emerges from flow matching methods, which aim to learn direct paths from noise to data. Methods like MeanFlow (Geng et al., 2025) approximate average velocity fields to enable one-step generation but rely on noise-sensitive Jacobian-vector products that can introduce gradient estimation errors, yet the model is not designed for mixed-type data generation. These errors propagate into synthetic data, blurring fine-grained features and weakening categorical relationships—a critical limitation in tabular domains where preserving precise feature interactions is essential. And the prevailing approach in one-step generation learns mean velocity fields along linear paths (Geng et al., 2025), using time warping functions to approximate the noise-to-data transformation. However, this linear path assumption fundamentally conflicts with tabular data's inherent geometric structure. Tabular feature spaces typically exhibit complex manifolds with heterogeneous densities—sharp categorical boundaries coexist with smooth continuous distributions. Linear trajectories inevitably traverse low-density regions, generating unrealistic feature combinations and degrading data quality. While manifold-aware learning has shown promise in computer vision (Gemici et al., 2016; Mathieu et al., 2019), its application to tabular data remains largely unexplored. ExpoTab addresses these limitations by learning geodesic paths on a structured manifold, providing stable one-step generation without compromising feature fidelity.

## 3 METHODOLOGY

This work introduces **ExpoTab**, a one-step generative framework for mixed-type tabular data that integrates manifold learning with flow matching. As illustrated in Figure 2, our approach consists of two core components: (1) ExpoTab-Encoder: a unified bounded representation space constructed via spherical embeddings for categorical features and normalized continuous representations (Figure 1), and (2) ExpoTab-Sampler: a manifold learning framework (via geodesic representation) that models the generative process by learning optimized trajectories as exponential maps on the data manifold (Figure 2B).

The ExpoTab framework reformulates the generative process by focusing on the geometry of the path rather than the velocity field. Conventional flow-matching models learn a time-dependent velocity field $v(z_t, t)$, where $z_t$ denotes a position in the background space at time $t$. Samples are then generated via numerical integration: $z_1 = z_0 + \int_0^1 v(z_t, t)dt$. In contrast, our framework enables direct, one-step generation by learning the intrinsic geometric properties of the generative trajectory itself.

The proposed approach begins by establishing a fixed background geometry using a unified encoder ('ExpoTab-Encoder'). We then posit that the true generative path from noise to data is a geodesic on an *implicit, learned* sub-manifold in the background space, as shown in Figure 2B. Instead of learning the complex, time-varying velocity field (tangent field) of these paths, we reframe the problem as a constant velocity mapping happening in a curved space, where the varying curvature accounts for all the acceleration. The core of our method is a neural network, $f_\theta$, trained to approximate the final outcome of the exponential map on this implicit geometry. It learns the total non-linear displacement caused by the path's curvature, allowing the final sample to be computed in a single step:

$$z_1 = z_0 + v_0 + f_\theta(z_0, v_0) \tag{1}$$

where $z_0 \in \mathcal{M}$ is an initial point on the implicit manifold, which shall be introduced later, and $v_0 = \mathbf{v}_\phi(z_0, 0)$ as obtained by flow-matching.

To provide the necessary supervision for this task, we employ a separate flow-matching network to compute the ground-truth displacement targets. This two-stage approach allows ExpoTab to learn the complex, path-integrated geometry efficiently, resulting in a principled and computationally efficient one-step generator.

### 3.1 THE DATA MANIFOLD $\mathcal{M}$ VIA EXPOTAB-ENCODER

The first step in our methodology is to establish a fixed data space. There are two major challenges: Accuracy and Scalability. To address those challenges, we propose ExpoTab-Encoder, which constructs a continuous and geometrically meaningful background data manifold, $\mathcal{M}_{bg}$, such that flow matching performs at a better accuracy and maintains its better separability when datasets scales larger, compared to existing methods (Figure 3). A key feature of ExpoTab-Encoder is its method for addressing the problem of *biased distribution of spherical embeddings*, where category points can cluster around the poles of a sphere. ExpoTab-Encoder mitigates this by leveraging a *Generalized Spiral* layout (detailed in Algorithm 1, lines 11-13) to ensure a uniform distribution of category points on the unit sphere, thereby maximizing inter-category separation.

ExpoTab-Encoder combines a spherical embedding for categorical features with a QuantileTransformer for numerical features (Figure 1). For continuous features, a Quantile Transformer $Q : \mathbb{R}^{D_c} \to [-1, 1]^{D_c}$ is applied to normalize the $D_c$ features. The resulting vector $\mathbf{x} = Q(\mathbf{x}_{\text{raw}})$ resides in a hypercube, a subset of the Euclidean space $\mathbb{R}^{D_c}$. For each of the $D_d$ categorical features, a deterministic mapping function $S_j : \{1, ..., K_j\} \to \mathbb{S}^2$ embeds each of the $K_j$ unique categories onto the surface of a 2-sphere Figure 2A. This process is implemented by the Encoder Procedure in Algorithm 1, which uses the generalized spiral parameterization to compute a unique 3D point for each category. A single encoded data point $\mathbf{z}_0$ is a tuple containing the encoded continuous and categorical vectors, forming the final background product manifold, $\mathcal{M}_{bg} = \mathbb{R}^{D_c} \times \prod_{j=1}^{D_d} \mathbb{S}^2$. As shown in Table 1, the designed space allows better performance for flow matching, which then can be further treated as a curve in the space of distribution that links the source and target distribution as two points(Figure 2B(a)(b)).

### 3.2 ONE-STEP GENERATION AS A LEARNED EXPONENTIAL MAP

In differential geometry, the **exponential map**, $\exp_z(\mathbf{v})$, is the operator that maps an initial point $z$ and a velocity vector $\mathbf{v}$ to the endpoint of the corresponding geodesic after a unit time. Calculating this map requires first knowing the manifold's geometry and then solving the geodesic equation, a process that is typically computationally intensive(Figure 2B(c)).

Our methodology takes a different approach. Instead of explicitly learning the geometry and then integrating, we train a single neural network, $f_\theta$, to directly approximate the *final outcome* of the exponential map. The geometry remains implicit, defined only by the single learned function $f_\theta$ that dictates the geodesic paths. The $k$-th component of this displacement vector approximates the double integral of the $k$-th component of the acceleration vector $\alpha_s$ at time $s$:

$$\int_0^1 \int_0^t \alpha_s^k ds dt = \int_0^1 \int_0^t \left( -\sum_{i,j} \Gamma_{ij}^k(s) v^i(s) v^j(s) \right) ds dt \tag{2}$$

$$\approx (f_\theta(\mathbf{z}_0, \mathbf{v}_0))^k \tag{3}$$

The network takes only the initial conditions of the path—the starting point $\mathbf{z}_1$ and initial velocity $\mathbf{v}_0$—and outputs a single displacement vector. This formulation allows generation to be performed in a single computational step. We computes the final point on the geodesic, $\mathbf{z}_0$, by adding the linear displacement from the initial velocity and the non-linear geometric displacement learned by the network to the starting point:

$$\exp_{z_0}(v_0) = z_0 + v_0 + f_\theta(z_0, v_0) \tag{4}$$

This approach provides the endpoint of a geodesic path in a single evaluation by abstracting the complex integration into the single learned function $f_\theta$.

#### TRAINING VIA FLOW MATCHING ON A HYPERBOLIC PATH

To provide a training signal for our one-step generator, we employ a conditional flow-matching framework. This involves a separate neural network, $\mathbf{v}_\phi(\mathbf{z}_t, t)$, trained to model the velocity field of a simple, predefined path between a noise point $z_0$ and a point in the dataset $z_1$. We define the path $\{z_t\}_t$ using a hyperbolic schedule, $g(t)$, which provides a beneficial inductive bias by mimicking

geodesic flow:

$$z_t = (1 - g(t))z_0 + g(t)z_1 \quad \text{where} \quad g(t) = \frac{\cosh(t) - 1}{\cosh(1) - 1} \tag{5}$$

The training and sampling procedures are detailed in Algorithms 8 and 9, with comprehensive implementation details provided in Appendix C.5. Besides, the comparison with other schedules is detailed in the extensive ablation study(see details in Appendix G.1).

EXPLICIT FORMULATION OF THE LEARNING OBJECTIVE

The learning objective for our main generator network, $f_\theta$, is to predict the displacement led by the curved space. This is framed as a regression problem where the target is derived from the flow-matching network's output:

$$\mathcal{L}(\theta) = \mathbb{E}_{z_0, z_1} \left[ \left\| f_\theta(z_0, v_0) - \left( \int_0^1 v_\phi(z_s, s)ds - v_0 \right) \right\|_2^2 \right] \tag{6}$$

Here, $v_\phi$ is the velocity field. By minimizing $\mathcal{L}$, the network $f_\theta$ learns the complex, non-linear component of the geodesic path, enabling efficient one-step generation.

## 4 EXPERIMENTAL SETTINGS

**Datasets.** We evaluate our model on diverse tabular datasets spanning mixed-type (categorical and continuous) and high-dimensional scenarios. Our benchmark includes 10 datasets from UCI and OpenML Repository: Adult, Default, Shoppers, Diabetes, Magic, Beijing, Bioresponse, Isolet, Gina, and Har. Statistics and preprocessing details are provided in Appendix D.1.

**Baselines.** We compare ExpoTab against state-of-the-art methods across two key dimensions: data encoding strategies and generative modeling approaches. For data encoding, we evaluate five representation schemes: One-hot encoding, Analog Bits encoding Chen et al. (2022), Dictionary encoding Si et al. (2025), Learned Embeddings Mikolov et al. (2013), and TabRep Si et al. (2025). These comparisons assess the effectiveness of our geometric embedding approach for handling mixed-type tabular data. For generative modeling, we benchmark against ten SOTA methods spanning diverse paradigms: classical oversampling (SMOTE Chawla et al. (2002)), GAN-based approaches (CTGAN Zhao et al. (2024)), diffusion models (TabDDPM Kotelnikov et al. (2023), TabSyn Zhang et al. (2023), CDTD Mueller et al. (2023), TabDiff Shi et al. (2024), TabRep-DDPM Si et al. (2025)), and flow matching methods (TabRep-Flow Si et al. (2025)). Additionally, we include Tab-GeoFlow and Tab-GeoDiff, which utilize our ExpoTab encoder with flow matching and diffusion backbones respectively, to isolate the contribution of our geometric encoding. See Appendix D.2 for implementation details and C for algorithmic specifications.

**Evaluation.** For the data representation, we assess encoder separability by evaluating prediction accuracy of a classifier trained and tested on synthetic data with for increasing-cardinality categorical features under Gaussian noise ($\sigma = 0, 0.01, 0.05, 0.1$). For the data generation, we assess synthetic data quality across three dimensions: (1) Fidelity ($\alpha$-precision, $\beta$-recall, column-wise distribution error, pairwise correlation error, C2ST); (2) Utility (Machine Learning Efficiency); and (3) Privacy (Distance to Closest Record (DCR)). See Appendix D.3 for details for all settings.

**Experimental Setup.** For the representation tasks, we first compare classifier prediction accuracy to evaluate the separability and scalability of the encoder. For a fair comparison, the Analog Bits encoding dimension is fixed at 3. Additionally, we evaluate how different encoding methods affect flow matching performance in downstream tasks, where Analog Bits encoding follows its original dimension-increasing scheme. While we exclude one-hot encoding from encoder comparisons due to dimensional explosion with high-cardinality data, we include it in flow matching evaluations by combining it with the Quantile Transformer. For the generation tasks, we evaluate ExpoTab against state-of-the-art generative models, with a focus on scalability, fidelity, and efficiency across data volumes from 100k to 1M samples. And we further assess the utility and privacy preservation of the generated data. Implementation details are provided in Appendix E.

## 5 Experiment Results

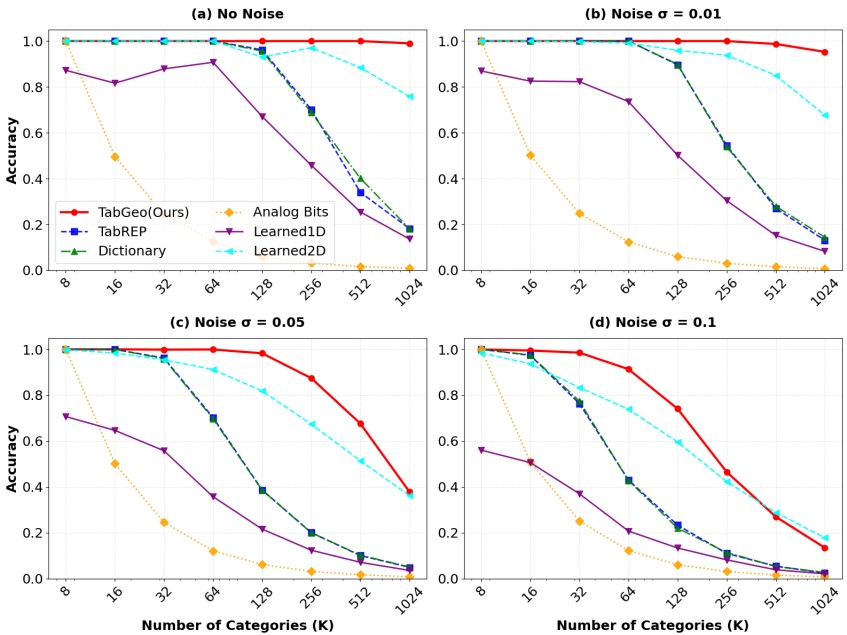

Figure 3: Comparison of separability and scalability for existing encoding methods

### 5.1 Encoder Separability

As shown in Figure 3, ExpoTab-Encoder achieve the highest performance across all experimental conditions. Compared to other advanced encoding methods, ExpoTab-Encoder demonstrates superior performance as noise levels increase from 0 to 0.1 and cardinalities scale from 8 to 1024. These results indicate that ExpoTab-Encoder achieves state-of-the-art separability for large category volumes while maintaining high scalability when processing high-cardinality data under noisy conditions.

### 5.2 Data representation for flow matching

Table 1: Comparison of different data representations schema for flow matching on Error Rate(%) of Shape and Machine Learning Efficiency where Adult, Default, Shoppers using AUC and Diabetes using the F1.

| Data Fidelity | Methods | Adult | Default | Shoppers | Diabetes |
|---|---|---|---|---|---|
| Error Rate(%) of Shape($\downarrow$) | OneHot-Flow | $9.34_{\pm0.09}$ | $8.05_{\pm0.08}$ | $7.16_{\pm0.10}$ | $3.35_{\pm0.06}$ |
| | Analog bits-Flow | $1.77_{\pm0.08}$ | $2.35_{\pm0.11}$ | $2.39_{\pm0.12}$ | $1.08_{\pm0.03}$ |
| | Dictionary-Flow | $1.90_{\pm0.05}$ | $2.73_{\pm0.04}$ | $3.11_{\pm0.11}$ | $0.99_{\pm0.04}$ |
| | TabRep-Flow | $1.44_{\pm0.03}$ | $2.63_{\pm0.06}$ | $2.38_{\pm0.04}$ | $1.00_{\pm0.04}$ |
| | Learned2D-Flow | $37.63_{\pm3.15}$ | $33.67_{\pm4.78}$ | $28.07_{\pm1.06}$ | $31.61_{\pm9.33}$ |
| | Tab-GeoFlow (Ours) | $\mathbf{1.11}_{\pm0.07}$ | $\mathbf{1.87}_{\pm0.05}$ | $\mathbf{2.25}_{\pm0.10}$ | $\mathbf{0.93}_{\pm0.02}$ |
| Data Utility | Methods | Adult | Default | Shoppers | Diabetes |
| Machine Learning Efficiency($\uparrow$) | OneHot-Flow | $0.895_{\pm0.007}$ | $0.759_{\pm0.005}$ | $0.910_{\pm0.006}$ | $0.372_{\pm.005}$ |
| | Analog bits-Flow | $0.911_{\pm0.001}$ | $0.763_{\pm0.004}$ | $0.910_{\pm0.005}$ | $0.372_{\pm.003}$ |
| | Dictionary-Flow | $0.910_{\pm0.002}$ | $0.759_{\pm.007}$ | $0.903_{\pm.006}$ | $0.376_{\pm.007}$ |
| | TabRep-Flow | $0.912_{\pm.004}$ | $0.756_{\pm.005}$ | $0.910_{\pm.005}$ | $0.373_{\pm.005}$ |
| | Learned2D-Flow | $0.126_{\pm.017}$ | $0.709_{\pm.009}$ | $0.868_{\pm.007}$ | $0.177_{\pm.008}$ |
| | Tab-GeoFlow (Ours) | $\mathbf{0.912}_{\pm.003}$ | $\mathbf{0.765}_{\pm.005}$ | $\mathbf{0.917}_{\pm.005}$ | $\mathbf{0.618}_{\pm0.021}$ |

As shown in Table 1, flow matching using ExpoTab-encoder consistently outperforms other representation methods on both error rate of shape and machine learning efficiency across mixed-type

Table 2: Comparison of generative models: sampling time with $\alpha$-precision shown in parentheses. **black** highlights the fastest sampling time. ExpoTab consistently achieves the fastest sampling times (red) while maintaining competitive $\alpha$-precision relative to the best-performing methods (blue).

| METHODS | SAMPLING TIME (S) ↓ ($\alpha$-PRECISION ↑) | | | | |
| --- | --- | --- | --- | --- | --- |
| | ADULT | BEIJING | DEFAULT | DIABETES | SHOPPERS |
| CTGAN | 0.86 (77.74) | 0.93 (96.27) | 1.08 (62.08) | 10.08 (79.89) | 0.41 (76.97) |
| TABDDPM | 44.96 (96.36) | 40.01 (97.93) | 32.46 (97.59) | 84.36 (28.35) | 23.22 (88.55) |
| TABSYN | 2.43 (99.39) | 2.56 (97.51) | 2.42 (98.65) | 7.89 (96.61) | 1.00 (98.36) |
| CDTD | 5.43 (99.37) | 3.53 (99.29) | 5.19 (99.36) | 21.89 (98.72) | 2.02 (98.89) |
| TABDIFF | 10.27 (99.02) | 6.86 (98.06) | 7.26 (98.49) | 29.23 (95.69) | 4.65 (99.11) |
| TABREP-FLOW | 1.52 (98.21) | 1.33 (98.16) | 1.01 (96.50) | 2.77 (99.08) | 0.50 (95.85) |
| TABREP-DDPM | 35.11 (99.11) | 26.44 (98.98) | 21.14 (98.66) | 54.84 (97.19) | 10.33 (96.14) |
| TAB-GEOFLOW | 2.89 (99.01) | 2.61 (98.91) | 1.97 (98.03) | 5.44 (99.15) | 0.93 (99.07) |
| TAB-GEODIFF | 26.64 (99.41) | 26.12 (99.05) | 21.31 (99.21) | 54.99 (97.79) | 10.64 (99.15) |
| **EXPOTAB (OURS)** | **0.20** (99.23) | **0.22** (98.47) | **0.21** (98.58) | **0.36** (98.16) | **0.19** (99.12) |
| SPEED IMPROVEMENT | **4.3×** ↑ | **4.2×** ↑ | **4.8×** ↑ | **7.7×** ↑ | **2.6×** ↑ |

of data. These results demonstrate that our geometric embedding approach provides more effective representations for flow matching, offering distribution preservation and enhanced utility for downstream tasks compared to other encoding schemes. See Tables 10 and 11 for more results.

## 5.3 SCALABILITY VS FIDEILITY VS SAMPLING SPEED

Figure 4A demonstrates ExpoTab's superior scalability compared to iterative generation methods on the Diabetes dataset. Experimental results show diffusion models (e.g., Tabsyn) require exponentially increasing time (1s→10,000s from 10k to 10M records) and flow matching methods show better but still significant scaling (0.2s→60s), while ExpoTab maintains near-linear O(n) scaling (0.19s→4.2s). This efficiency stems from our one-step generation paradigm, avoiding iterative refinement costs. The fidelity-efficiency trade-off analysis (Figure 4B) reveals ExpoTab's optimal balance. This makes ExpoTab particularly valuable for large-scale applications where both accuracy and throughput matter.

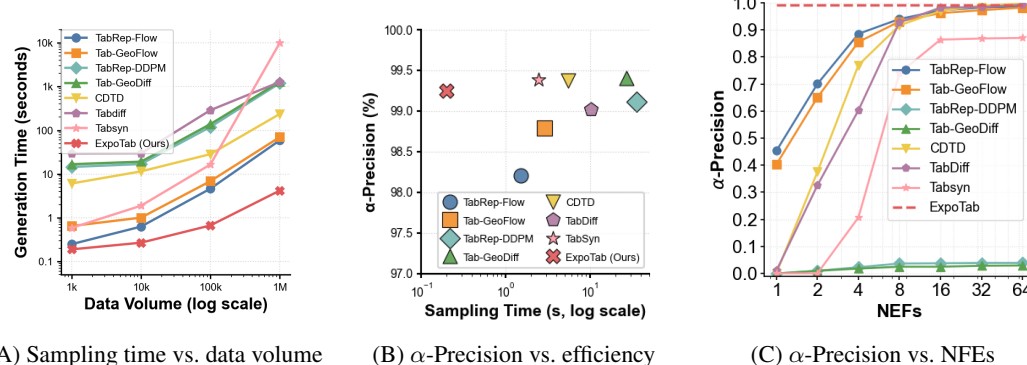

(A) Sampling time vs. data volume    (B) $\alpha$-Precision vs. efficiency    (C) $\alpha$-Precision vs. NFEs

Figure 4: Comparative performance of tabular data generation methods: (A) Sampling time versus data volume shows ExpoTab maintains stable performance at scale; (B) Trade-off between $\alpha$-Precision and generation efficiency; (C) NFEs-Fidelity tradeoff.

## 5.4 PERFORMANCE COMPARED TO MULTI-STEP GENERATIVE MODELS.

**Quality vs Efficiency.** As shown in Table 2, ExpoTab exceeds the sampling speed of the current fastest models—GAN-based and flow-matching-based methods—by a factor of 2.6 to 7.7. Further-

more, ExpoTab achieves 99.23% $\alpha$-precision on the Adult dataset, which is merely 0.18% below the state-of-the-art method (Tab-GeoDiff). Notably, it can generate over 30,000 data points in 0.2 seconds—demonstrating a $133\times$ speedup compared to Tab-GeoDiff while maintaining comparable fidelity. This significant efficiency gain highlights how ExpoTab substantially reduces the computational overhead of conventional iterative methods while preserving data quality.

**Utility vs Privacy.** We further validate ExpoTab's synthetic data quality by evaluating both downstream machine learning performance (AUC) and privacy protection (DCR) on high-dimensional datasets. The results show that ExpoTab achieves competitive machine learning efficacy, matching or exceeding state-of-the-art AUC scores while maintaining competitive privacy protection. See the reults in Appendix H.

## 5.5 Ablation Studies

**NEFs vs Data Fidelity** We evaluate ExpoTab's performance across function evaluations (1 to 64 NFEs) compared to state-of-the-art methods on the Adult dataset. As shown in Figure 4C, ExpoTab achieves superior convergence efficiency, reaching competitive $\alpha$-precision in just one step. This contrasts with existing methods that suffer significant degradation under single-step constraints. Furthermore, achieving comparable performance requires 64 or more sampling steps for many existing methods, such as TabDiff, CDTD, TabREP-Flow, and Tab-GeoFlow. Other methods, including TabSyn, TabRep-DDPM, and Tab-GeoDiff, fail to reach a comparable $\alpha$-precision even when using 64 steps. Furthermore, additional ablation studies in Appendix G demonstrate that optimal configurations of key components—including time sampling strategies, path warping functions, and classifier-free guidance scale—are important for achieving peak performance with ExpoTab.

## 6 Conclusions

To address the challenges of efficient high-fidelity tabular data generation, we introduced **ExpoTab**, a novel one-step generative framework specifically designed for mixed-type data. ExpoTab unifies high-cardinality categorical features and continuous features within a bounded representation space by combining spherical embeddings with quantile transformations. Building on this representation, ExpoTab models generation as adaptive geodesic trajectories on the manifold, directly approximating the exponential map to achieve one-step sampling that balances fidelity and efficiency. Extensive experiments demonstrate that ExpoTab delivers fast, scalable generation (up to 0.1s for 1M samples) while maintaining strong performance on fidelity, utility, and privacy across diverse benchmarks. Nevertheless, our current one-step formulation primarily captures principal flow directions and may overlook fine-grained feature distributions, leading to gaps in certain column-wise and pairwise fidelity metrics when compared to state-of-the-art multi-step generative models. Future work will focus on enhancing ExpoTab's ability to jointly model global and local data characteristics without sacrificing computational efficiency, further strengthening its role as a practical solution for large-scale tabular data synthesis.

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

# Appendix

## CONTENTS

## A  FLOW MATCHING

### A.1  FLOW MATCHING AND AVERAGE VELOCITY

Flow Matching (Lipman et al., 2023) learns a velocity field $\mathbf{v}_\theta(\mathbf{z}_t, t)$ that transports samples from a noise distribution to the data distribution. The Flow Matching objective is defined as:

$$\mathcal{L}_{\text{FM}}(\theta) = \mathbb{E}_{t, p_t(\mathbf{z}_t)} \left| \left| \mathbf{v}_\theta(\mathbf{z}_t, t) - \mathbf{v}(\mathbf{z}_t, t) \right| \right|^2. \tag{7}$$

Samples are generated by solving the ordinary differential equation:

$$\frac{d}{dt}\mathbf{z}_t = \mathbf{v}_\theta(\mathbf{z}_t, t), \quad \mathbf{z}_1 \sim \mathcal{N}(0, I). \tag{8}$$

Rather than modeling instantaneous velocity, the *average velocity* $\mathbf{u}$ (Geng et al., 2025) represents the mean displacement between two time steps:

$$\mathbf{u}(\mathbf{z}_t, r, t) = \frac{1}{t - r} \int_r^t \mathbf{v}(\mathbf{z}_\tau, \tau) \, d\tau. \tag{9}$$

Learning $\mathbf{u}$ enables accurate one-step generation by directly predicting the next transformation from noise to data, avoiding iterative refinement.

## B  DIFFERENTIAL GEOMETRY PRELIMINARIES

### B.1  MANIFOLDS

**Definition 1** (Smooth Manifold). *A **smooth manifold**, $\mathcal{M}$, is a space that, on a local scale, resembles Euclidean space ($\mathbb{R}^n$). More formally, it is a topological space for which every point has a neighbourhood that is diffeomorphic to an open subset of $\mathbb{R}^n$. The integer $n$ is the dimension of the manifold.*

Intuitively, a manifold can be a curved object, but if you "zoom in" on any single point, its immediate surroundings look flat. A classic example is the surface of a sphere, $Sphere^2$. While globally it is curved, any small patch of the sphere is well-approximated by a flat plane ($\mathbb{R}^2$).

In our work, we model the space of tabular data as a manifold. This allows us to handle the complex, non-linear relationships between features in a geometrically principled way. Specifically, we construct a **background data manifold**, $\mathcal{M}_{bg}$, as the Cartesian product of the spaces for the numerical and categorical features:

$$\mathcal{M}_{bg} = \mathbb{R}^{D_c} \times \prod_{j=1}^{D_d} Sphere^2$$

Here, each of the $D_d$ categorical features is mapped to its own 2-sphere ($Sphere^2$), and the $D_c$ continuous features reside in a Euclidean space. This product space is itself a smooth manifold, providing a fixed geometric background on which our generative process operates.

### B.2  TANGENT SPACES AND VECTORS

**Definition 2** (Tangent Space). *For each point $p$ on a smooth manifold $\mathcal{M}$, there is an associated vector space, $T_p\mathcal{M}$, called the **tangent space** at $p$. The tangent space is the set of all possible velocity vectors of smooth curves passing through $p$. It can be thought of as the best linear approximation of the manifold around the point $p$.*

The dimension of the tangent space $T_p\mathcal{M}$ is equal to the dimension of the manifold $\mathcal{M}$. Vectors in this space are called **tangent vectors**.

This concept is crucial for understanding the dynamics of our model. In our framework, the generative path from a noise point $\mathbf{z}_1$ to a data point $\mathbf{z}_0$ is a curve on an implicit, learned manifold. The

process is initiated with a starting point $\mathbf{z}_1 \in \mathcal{M}$ and an **initial velocity** $\mathbf{v}_0$. This initial velocity is a tangent vector that resides in the tangent space at the starting point, formally written as:

$$\mathbf{v}_0 \in T_{\mathbf{z}_1}\mathcal{M}$$

The vector $\mathbf{v}_0$ defines the initial direction of the geodesic path on the manifold, analogous to specifying a direction in which to travel from a point in Euclidean space.

### B.3 RIEMANNIAN METRIC

**Definition 3** (Riemannian Metric). *A **Riemannian metric**, g, on a smooth manifold $\mathcal{M}$ is a smooth assignment of an inner product, $g_p(\cdot, \cdot)$ or $\langle \cdot, \cdot \rangle_p$, to each tangent space $T_p\mathcal{M}$. A manifold endowed with such a metric is called a **Riemannian manifold**.*

The metric allows us to perform geometric measurements on the manifold. For any two tangent vectors $\mathbf{u}, \mathbf{v} \in T_p\mathcal{M}$, the metric can be used to define:

- **Length of a vector:** $\|\mathbf{v}\|_p = \sqrt{g_p(\mathbf{v}, \mathbf{v})}$.

- **Angle between vectors:** The angle $\theta$ between $\mathbf{u}$ and $\mathbf{v}$ is given by $\cos\theta = \frac{g_p(\mathbf{u}, \mathbf{v})}{\|\mathbf{u}\|_p \|\mathbf{v}\|_p}$.

- **Length of a curve:** The length of a curve $\gamma : [a, b] \to \mathcal{M}$ is found by integrating the speed: $L(\gamma) = \int_a^b \|\dot{\gamma}(t)\|_{\gamma(t)} dt$.

While our background manifold $\mathcal{M}_{bg}$ has a simple, known product metric, the core idea of ExpoTab is that the generative path exists on an *implicit* manifold with a more complex, unkown metric. The geometry induced by this unknown metric is what our network $f_\theta$ learns to navigate.

### B.4 CHRISTOFFEL SYMBOLS

In a curved space, the basis vectors used to describe tangent vectors change as we move from one point to another. The Christoffel symbols are the coefficients that precisely quantify this change.

**Definition 4** (Christoffel Symbols). *The **Christoffel symbols of the second kind**, denoted $\Gamma_{ij}^k$, are a set of real-valued functions on the manifold that describe the covariant derivative (a generalization of the directional derivative) in local coordinates. They encapsulate the properties of the Riemannian metric and its derivatives.*

Essentially, the Christoffel symbols encode all the intrinsic geometric information of the manifold, including its \*\*curvature\*\*. They are the crucial components in the equation for geodesic paths, as they define what it means for a path to be "straight." As we will see in the next section, the acceleration of a curve on a manifold is expressed using these symbols. The term $-\sum_{i,j} \Gamma_{ij}^k v^i v^j$ from our main paper represents exactly this geometric acceleration, which our model learns to integrate.

### B.5 GEODESICS AND THE GEODESIC EQUATION

**Definition 5** (Geodesic). *A **geodesic** is a curve $\gamma(t)$ on a manifold that is locally distance-minimizing. It is the generalization of a straight line in Euclidean space. Formally, a geodesic is a curve whose tangent vector remains parallel to itself as it is transported along the curve. This is equivalent to saying the curve has **zero acceleration** on the manifold.*

In Euclidean space, a curve with zero acceleration is a straight line. On a manifold, defining acceleration is more complex due to curvature. The acceleration of a curve $\gamma(t)$ is given by its **covariant derivative** along itself, $\nabla_{\dot{\gamma}}\dot{\gamma}$. A geodesic is a curve for which this acceleration is zero. In local coordinates, this condition translates to a system of second-order differential equations.

**Definition 6** (Geodesic Equation). *A curve $\gamma(t)$ is a geodesic if its components satisfy the **geodesic equation** for each component $k$:*

$$\frac{d^2\gamma^k}{dt^2} + \sum_{i,j} \Gamma_{ij}^k(\gamma(t)) \frac{d\gamma^i}{dt} \frac{d\gamma^j}{dt} = 0$$

*where $\Gamma_{ij}^k$ are the Christoffel symbols and $\frac{d\gamma^i}{dt}$ are the components of the velocity vector $\dot{\gamma}(t)$.*

The first term, $\frac{d^2\gamma^k}{dt^2}$, is the familiar component-wise acceleration from calculus. The second term, involving the Christoffel symbols, is the correction term that accounts for the curvature of the manifold.

### B.5.1 CONNECTION TO EXPOTAB

This equation is directly related to the methodology of ExpoTab. In our paper, we define the **geodesic acceleration vector**, $\alpha_s$, whose components are given by:

$$\alpha_s^k = -\sum_{i,j} \Gamma_{ij}^k(s)v^i(s)v^j(s)$$

where $v^i(s)$ are the components of the velocity at time $s$. By comparing this with the geodesic equation, we see that $\alpha_s^k$ is precisely the geometric contribution to the curve's acceleration. A path is a geodesic if its ordinary acceleration is exactly canceled out by this geometric acceleration term.

The core of our method is to learn the total displacement caused by this geodesic acceleration. The network $f_\theta$ is trained to approximate the double integral of this acceleration over the entire path:

$$(f_\theta(\mathbf{z}_1, \mathbf{v}_0))^k \approx \int_0^1 \int_0^t \alpha_s^k ds dt$$

By doing so, $f_\theta$ learns to predict the total non-linear deviation from a straight Euclidean line that is required to stay on the "straight" geodesic path of the underlying learned manifold.

### B.6 THE EXPONENTIAL MAP

**Definition 7** (Exponential Map). *The **exponential map**, denoted $\exp_p$, maps a tangent vector at a point $p$ to a point on the manifold $\mathcal{M}$. It is defined by following the geodesic that starts at $p$ in the direction of the tangent vector. Specifically, for a tangent vector $\mathbf{v} \in T_p\mathcal{M}$, $\exp_p(\mathbf{v})$ is the point $\gamma(1)$ on the unique geodesic $\gamma(t)$ that satisfies the initial conditions $\gamma(0) = p$ and $\dot{\gamma}(0) = \mathbf{v}$.*

In simpler terms, the exponential map "shoots" you from a starting point $p$ along a geodesic for a specified direction and "distance" (determined by the vector $\mathbf{v}$). It is the direct generalization of vector addition in Euclidean space. In $\mathbb{R}^n$, moving from a point $p$ with velocity $v$ for one unit of time results in the endpoint $p + v$. On a manifold, this operation is replaced by $\exp_p(\mathbf{v})$.

### B.6.1 CONNECTION TO THE EXPOTAB GENERATOR

The exponential map provides the formal geometric interpretation of our one-step generator, $G_\theta$. The core equation of our generator is:

$$G_\theta(\mathbf{z}_1, \mathbf{v}_0) = \mathbf{z}_1 + \mathbf{v}_0 + f_\theta(\mathbf{z}_1, \mathbf{v}_0)$$

We can interpret this formula as a learned approximation of the exponential map on the implicit manifold:

$$\exp_{\mathbf{z}_1}(\mathbf{v}_0) \approx \mathbf{z}_1 + \mathbf{v}_0 + f_\theta(\mathbf{z}_1, \mathbf{v}_0)$$

Here:

- The term $\mathbf{z}_1 + \mathbf{v}_0$ represents a step along a straight line in the ambient Euclidean space. This is what the endpoint would be if the manifold had no curvature.

- The network $f_\theta(\mathbf{z}_1, \mathbf{v}_0)$ learns the precise correction vector needed to account for the manifold's curvature. It predicts the total deviation between the simple Euclidean path and the true geodesic path.

Thus, our entire generator $G_\theta$ acts as a learned approximation of the exponential map, $G_\theta \approx \exp$. By training $f_\theta$ to predict the result of integrating the geodesic acceleration, we enable the model to perform this complex geometric operation in a single, efficient forward pass.

## C DETAILED ALGORITHMS

### C.1 EXPOTAB-ENCODER ENCODER/DECODER

The ExpoTab-Encoder framework relies on a novel spherical encoding scheme to map discrete categorical values onto a continuous, structured 3D space. This transformation is fundamental, as it allows subsequent continuous generative models (e.g., flow matching, diffusion) to operate natively on categorical data without resorting to standard one-hot encodings, which can be high-dimensional and lack geometric structure. The encoder and decoder procedures are detailed in Algorithm 1. The core idea is to represent each possible category for a feature as a unique point on the surface of the unit sphere $\mathbb{S}^2$. This is achieved using a spiral sampling method based on the golden angle. For a categorical feature $j$ with cardinality $K_j$, we generate $K_j$ points $\mathcal{P}^j = \{\mathbf{p}_0, \ldots, \mathbf{p}_{K_j-1}\}$. The spherical coordinates $(\theta_n, \phi_n)$ for each point are computed iteratively:

- The height $h_n$ is spaced uniformly between -1 and 1.
- The azimuth $\phi_n$ is incremented by the golden angle $\pi(3 - \sqrt{5})$ radians for each subsequent point.

This method ensures that the points are distributed in a visually uniform and well-separated pattern on the sphere, maximizing the minimum distance between any two points. This ensures that the encoded representations are geometrically distinct, which is crucial for stable training and accurate decoding.

The **Encoder** maps a batch of integer-valued categorical data $X$ to a batch of 3D coordinates $E$. For each feature and each data point, it simply retrieves the pre-computed spherical point corresponding to the category index. The output is a continuous tensor where each categorical value is represented by a 3D vector, effectively reducing the dimensionality from $K_j$ (in one-hot space) to 3 while preserving semantic distinctness through geometric distance.

The **Decoder** performs the inverse mapping from the continuous latent space back to discrete categories. For each encoded 3D vector, it finds the nearest point in the pre-computed set $\mathcal{P}^j$ for that feature. A threshold $\epsilon$ is applied to the vector's norm to handle potential invalid points that may be generated by the model, allowing for robustness to noise.

This spherical encoding provides several key advantages:

- **Structured Latent Space**: It imposes a meaningful geometric structure where semantic similarity can, in principle, be encoded by spatial proximity, a property absent in one-hot encodings.
- **Dimensionality Reduction**: It represents a category with only 3 dimensions instead of $K_j$, significantly reducing the parameter count for the subsequent generative model, especially for high-cardinality features.
- **Model Agnosticism**: The resulting continuous representations can be seamlessly used by any generative model designed for continuous data, such as normalizing flows or diffusion models.

### C.2 TAB-GEOFLOW

Our first proposed multi-step generative model, Tab-GeoFlow, employs continuous normalizing flows to learn a vector field $v_\theta(z_t)$ that matches a predefined conditional flow $u_t(z_t \mid z_0)$. The conditional flow matching (CFM) objective is to minimize the expected $L_2$-norm between the estimated and target velocity fields:

$$\mathcal{L}_{\text{CFM}} = \mathbb{E}_{z_0, z_1, \tilde{t}, \epsilon} \left[ \left\| f_\theta(z_t, \tilde{t}) - (z_1 - z_0) \right\|_2^2 \right], \tag{10}$$

where $f_\theta$ is the neural approximation of the vector field. This loss encourages the model to learn conditional flows that transport synthetic embeddings toward real ones across diverse tabular domains. Here, the conditional flow is defined as $u_t(z_t \mid z_0) = \epsilon - z_0$, which determines the direction of the flow. This formulation approximates the probability path $\mathcal{N}(z_t, \sigma_t^2 I)$, where the mean is a linear interpolation between $z_0$ and $z_1$, and the time-dependent standard deviation $\sigma_t$ controls the amount of Gaussian noise injected at each step.

Furthermore, we introduce an adaptive time warping mechanism to prioritize structurally critical regions of the transport path in flow matching. Given a uniformly sampled time $u \sim \mathcal{U}(0, 1)$, we apply the transformation $\tilde{t} =$

---

**Algorithm 1** Spherical Encoder and Decoder

---

1: **Input:**   For encoder: $X \in \mathbb{Z}^{B \times N}$ (categorical inputs), $\mathbf{K} = (K_1, \ldots, K_N)$ (feature cardinalities)
2:   For decoder: $E \in \mathbb{R}^{B \times 3N}$ (encoded data), $\mathbf{K}$, $\epsilon > 0$ (decoding threshold)
3: **Output:**   For encoder: $E \in \mathbb{R}^{B \times 3N}$ (spherical encodings)
4:   For decoder: $D \in \mathbb{Z}^{B \times N}$ (decoded values)
5: **Encoder Procedure:**
6: Initialize $E \leftarrow \mathbf{0}^{B \times 3N}$
7: **for** $j = 1$ **to** $N$ **do**
8:    **for** $n = 0$ **to** $K_j - 1$ **do**
9:       $h_n \leftarrow -1 + \frac{2n}{K_j - 1}$
10:       $\theta_n \leftarrow \arccos(h_n)$
11:       $\phi_n \leftarrow \begin{cases} 0 & \text{if } n = 0 \\ (\phi_{n-1} + \pi(3 - \sqrt{5})) \bmod 2\pi & \text{else} \end{cases}$
12:       $\mathbf{p}_n \leftarrow (\sin \theta_n \cos \phi_n, \sin \theta_n \sin \phi_n, \cos \theta_n)$
13:    **end for**
14:    Store $\mathcal{P}^j = \{\mathbf{p}_0, \ldots, \mathbf{p}_{K_j - 1}\}$
15:    **for** $b = 1$ **to** $B$ **do**
16:       $E[b, 3j - 3 : 3j] \leftarrow \mathbf{p}_{X_{bj}}$
17:    **end for**
18: **end for**
19: **Decoder Procedure:**
20: Initialize $D \leftarrow -1^{B \times N}$
21: **for** $j = 1$ **to** $N$ **do**
22:    Recompute $\mathcal{P}^j$ identically to encoder
23:    **for** $b = 1$ **to** $B$ **do**
24:       $\mathbf{e} \leftarrow E[b, 3j - 3 : 3j]$
25:       **if** $\|\mathbf{e}\|_2 \geq \epsilon$ **then**
26:          $D_{bj} \leftarrow \arg\min_k \|\mathbf{e} - \mathbf{p}_k\|_2$
27:       **end if**
28:    **end for**
29: **end for**
30: **return** appropriate output ($E$ for encoder, $D$ for decoder)

---

**Algorithm 2** Training

---

**Require:** Data distribution $p(z)$, model $h_\theta$
1: **while** not converged **do**
2:    Sample data $z_0 = [x_0, c_0] \sim p(z)$
3:    Encode categorical part $c_0 \leftarrow$ ExpoTab-Encoder($c_0, K$)
4:    Encode numerical part $x_0 \leftarrow$ QuantileTransformer($x_0$)
5:    $z_0 \leftarrow$ concat($x_0, c_0$)
6:    Sample time $t \sim$ Uniform($0, 1$)
7:    Sample noise $\epsilon \sim \mathcal{N}(0, I)$
8:    Compute noisy sample $z_t \leftarrow (1 - t)z_0 + t\epsilon$
9:    Define target velocity $u_t \leftarrow \epsilon - z_0$
10:   Calculate loss $\mathcal{L} = \|h_\theta(z_t, t) - u_t\|^2$
11:   Update $\theta$ using gradient descent on $\mathcal{L}$
12: **end while**

---

$\frac{\sqrt{u}}{1 + k\sqrt{u}}$, where $k$ is a tunable parameter controlling the warping intensity. This skew allocates higher gradient emphasis to initial steps, where small displacements drive significant structural changes, akin to curriculum learning. For $k = 0$, approximately 68% of samples lie in the first 50% of the trajectory, as $\mathbb{P}(\tilde{t} \leq 0.5) = \mathbb{P}(u \leq 0.25) = 0.25$. Fine-tuning $k$ allows flexibility: positive $k$ compresses the warping, while negative $k$ accelerates it. Empirical results in Experiments section demonstrate that $k = 0$ ($\tilde{t} = \sqrt{u}$) optimally balances efficiency and performance, matching or surpassing adaptive approaches with non-zero $k$, due to its simplicity, smooth derivative, and alignment with optimal transport dynamics.

---

**Algorithm 3** Sampling

---

**Require:** Trained model $v_\theta$, number of steps $T$
**Ensure:** Generated sample $z_0'$
1: Sample initial noise $z_T \sim \mathcal{N}(0, I)$
2: **for** $i \leftarrow T$ down to 1 **do**
3:    $z_{i-1} \leftarrow z_i + \frac{1}{T} \cdot v_\theta(z_i, i/T)$ {Euler integration step}
4: **end for**
5: Split $z_0 = [x_0, c_0]$
6: Decode $c_0' \leftarrow$ InvExpoTab-Encod$(c_0, K)$
7: Decode $x_0' \leftarrow$ InvQuantileTransformer$(x_0)$
8: $z_0' \leftarrow$ concat$(x_0', c_0')$
9: **return** $z_0'$

---

## C.3 TAB-GEODIFF

Our second proposed multi-step model, TabGeo-Diff, is a diffusion-based framework for mixed-type tabular data that combines DDPM-style training with geometric feature embeddings. The model employs a cosine noise schedule to gradually corrupt inputs through forward diffusion, learning to predict noise via score matching. For numerical features, we apply quantile transformers to ensure normalized distributions, while categorical variables are embedded on a hypersphere to maintain geometric relationships in a continuous latent space. The diffusion process incorporates adaptive mollification and Gaussian smoothing to ensure smooth feature transitions during generation. Unlike conventional approaches that require post-hoc sampling acceleration, TabGeo-Diff achieves efficient synthesis through its optimized noise schedule and spherical embedding space, preserving the topological structure of both continuous and discrete variables. The implementation details are shown in the Algorithm 4.

---

**Algorithm 4** Training Process

---

**Require:** Data $p(z_0)$, steps $T$, smoothness $\gamma$
1: {Define cosine noise schedule}
2: $\bar{\alpha}_t \leftarrow \cos^2\left(\frac{(t/T+\epsilon)}{1+\epsilon} \cdot \frac{\pi}{2}\right)$
3: $\alpha_t \leftarrow \sqrt{\bar{\alpha}_t / \bar{\alpha}_{t-1}}$
4: $\sigma_t \leftarrow \sqrt{1 - \alpha_t^2}$
5: **while** not converged **do**
6:    Sample $z_0 = [x_0, c_0] \sim p(z_0)$
7:    {Adaptive Smoothing}
8:    $\epsilon \leftarrow 0.01$
9:    **while** $\|x_0[:, 1 :] - x_0[:, : -1]\|_F > \gamma$ **do**
10:      $x_0 \leftarrow x_0 + \epsilon \cdot \mathcal{N}(0, I)$
11:      $\epsilon \leftarrow 1.5 \cdot \epsilon$
12:    **end while**
13:    $c_0 \leftarrow$ S$(c_0, K)$ {Spherical embedding}
14:    $x_0 \leftarrow$ Q$(x_0)$ {Quantile transformer}
15:    $z_0 \leftarrow$ concat$(x_0, c_0)$
16:    Sample $t \sim$ Uniform$(\{1, \dots, T\})$
17:    Sample noise $\epsilon \sim \mathcal{N}(0, I)$
18:    $z_t \leftarrow \sqrt{\bar{\alpha}_t} z_0 + \sqrt{1 - \bar{\alpha}_t} \epsilon$ {Forward diffusion}
19:    {Compute loss based on score matching}
20:    $\mathcal{L}(\theta) \leftarrow \left\| s_\theta(z_t, t) + \frac{\epsilon}{\sqrt{1 - \bar{\alpha}_t}} \right\|^2$
21:    Update $\theta$ via gradient descent on $\nabla_\theta \mathcal{L}(\theta)$
22: **end while**

---

---

**Algorithm 5** DDPM Sampling

---

**Require:** Trained model $\epsilon_\theta$, noise schedule $\{\alpha_t, \bar{\alpha}_t\}$, total steps $T$
1: $z_T \sim \mathcal{N}(0, I)$ {Sample initial noise}
2: **for** $t \leftarrow T$ down to 1 **do**
3:   $\epsilon \sim \mathcal{N}(0, I)$ **if** $t > 1$, **else** $\epsilon \leftarrow 0$
4:   Predict noise: $\hat{\epsilon}_t \leftarrow \epsilon_\theta(z_t, t)$
5:   Compute mean: $\mu_\theta(z_t, t) \leftarrow \frac{1}{\sqrt{\alpha_t}} \left( z_t - \frac{1-\alpha_t}{\sqrt{1-\bar{\alpha}_t}} \hat{\epsilon}_t \right)$
6:   $z_{t-1} \leftarrow \mu_\theta(z_t, t) + \sqrt{1 - \bar{\alpha}_{t-1}} \cdot \epsilon$ {Add noise back for previous step}
7: **end for**
8: Decode $z_0$: $c_0 \leftarrow \text{InvS}(z_0^c)$, $x_0 \leftarrow \text{InvQ}(z_0^x)$ {Revert preprocessing}
9: **return** $z_0 = [x_0, c_0]$

---

## C.4 TAB-MEANFLOW

The proposed **Tab-MeanFlow** framework applies the MeanFlow principle () to tabular data generation. This approach leverages a linear path schedule $g(t) = t$, which induces a straight trajectory in the latent space between noise $\mathbf{z}_1 \sim \mathcal{N}(0, \mathbf{I})$ and the data point $\mathbf{z}0$. The velocity model $f\theta$ is trained to approximate the constant flow along this straight path.

For training, we minimize an adaptively weighted $L_2$ loss $|\Delta|_2^2$, where the weight $w = (|\Delta|_2^2 + c)^{-p}$ (with $c = 10^{-3}$) introduces a pseudo-Huber-like regularization when $p = 0.5$, balancing sensitivity to outliers.

At inference (Algorithm 7), the one-step generation is achieved by simply reversing the learned linear flow: the model predicts the data point $\mathbf{z}_0 = \mathbf{z}1 - f\theta(\mathbf{z}_1)$, followed by the decoding of the categorical (InvTabGEO) and continuous (InvQuantileTransformer) features. This avoids the need for complex ODE solvers while preserving the theoretical guarantees of flow matching.

---

**Algorithm 6** Tab-MeanFlow Training

---

**Require:** Tabular dataset $\mathcal{D} = \{\mathbf{x}_i, \mathbf{c}_i\}_{i=1}^N$; Curvature exponent $k$; encoder TabGEO$_\phi$; velocity model $f_\theta$
1: Initialize parameters $\theta$, $\phi$
2: **while** not converged **do**
3:   Sample batch $\mathbf{z}_0 = [\mathbf{x}_0, \mathbf{c}_0] \sim \mathcal{D}$
4:   $\mathbf{c}_0 \leftarrow \text{TabGEO}_\phi(\mathbf{c}_0, K)$
5:   $\mathbf{x}_0 \leftarrow \text{QuantileTransformer}(\mathbf{x}_0)$
6:   $\mathbf{z}_0 \leftarrow \text{concat}(\mathbf{x}_0, \mathbf{c}_0)$
7:   Sample $t \sim \text{Uniform}(0, 1)$, then $t \leftarrow \sqrt{t}$ {Time warping}
8:   Sample $r \sim \text{Uniform}(0, 1)$ and $r \leftarrow r \cdot t$
9:   Sample noise $\boldsymbol{\epsilon} \sim \mathcal{N}(0, \mathbf{I})$
10:   $\mathbf{z}_1 \leftarrow \boldsymbol{\epsilon}$
11:   Compute path schedule $g(t) = t^k$, $g'(t) = k \cdot t^{k-1}$
12:   Interpolate on path: $\mathbf{z}_t \leftarrow (1 - g(t)) \cdot \mathbf{z}_1 + g(t) \cdot \mathbf{z}_0$
13:   Compute path velocity: $\mathbf{u} \leftarrow g'(t) \cdot (\mathbf{z}_0 - \mathbf{z}_1)$
14:   $\hat{\mathbf{u}}, \frac{d\hat{\mathbf{u}}}{dt} \leftarrow \text{JVP}(f_\theta, (\mathbf{z}_t, r, t), (\mathbf{u}, 0, 1))$ {Jacobian-Vector Product}
15:   $\mathbf{u}_{\text{target}} \leftarrow \mathbf{u} - (t - r) \cdot \frac{d\hat{\mathbf{u}}}{dt}$ {Compute corrected velocity target}
16:   $\mathcal{L} \leftarrow \|\hat{\mathbf{u}} - \text{stop\_gradient}(\mathbf{u}_{\text{target}})\|_2^2$
17:   Update $\theta$, $\phi$ using $\nabla_{\theta,\phi}\mathcal{L}$
18: **end while**

---

## C.5 EXPOTAB

We propose a unified approach based on the MeanFlow framework, which learns the average velocity field to enable accurate one-step generation. Unlike the two-stage method, this approach employs a *single* neural network $\mathbf{u}_\theta(\mathbf{z}_t, r, t)$ trained to predict the average velocity between times $r$ and $t$ for a non-linear probability path defined by a warp function $g(t)$.

---

**Algorithm 7** One-Step Sampling for TabGeo-MeanFlow

---

**Require:** Trained model $f_\theta$, feature cardinalities $K$
1: Sample $\mathbf{z}_1 \sim \mathcal{N}(0, I)$ {Sample base noise}
2: $\hat{\mathbf{u}} \leftarrow f_\theta(\mathbf{z}_1, t = 1)$ {Predict velocity at $t = 1$}
3: $\mathbf{z}_0 \leftarrow \mathbf{z}_1 - \hat{\mathbf{u}}$ {Reverse the flow in one step}
4: Split $\mathbf{z}_0 = [\mathbf{x}_0, \mathbf{c}_0]$
5: $\mathbf{c}_0 \leftarrow \text{InvTabGEO}(\mathbf{c}_0, K)$ {Decode categoricals}
6: $\mathbf{x}_0 \leftarrow \text{InvQuantileTransformer}(\mathbf{x}_0)$ {Decode continuous}
7: **return** $\mathbf{z}_0 = [\mathbf{x}_0, \mathbf{c}_0]$

---

---

**Algorithm 8** ExpoTab Training

---

**Require:** Dataset $\mathcal{D}$, encoder TabGEO$_\phi$, velocity model $\mathbf{v}_\phi$, displacement model $f_\theta$, path $g(t)$
1: Initialize parameters $\phi, \theta$
2: **Stage 1: Train the Velocity Field $\mathbf{v}_\phi$**
3: **while** $\mathbf{v}_\phi$ not converged **do**
4:     Sample $\mathbf{z}_1 \sim \mathcal{D}$, encode: $\mathbf{z}_1 \leftarrow \text{Encode}(\mathbf{z}_1)$
5:     Sample $\mathbf{z}_0 \sim \mathcal{N}(0, I)$
6:     Sample $t \sim U(0, 1)$
7:     $\mathbf{z}_t \leftarrow (1 - g(t))\mathbf{z}_0 + g(t)\mathbf{z}_1$ {Interpolate on hyperbolic path}
8:     $\mathbf{u}_{\text{target}} \leftarrow g'(t)(\mathbf{z}_1 - \mathbf{z}_0)$ {Instantaneous velocity target}
9:     $\hat{\mathbf{u}} \leftarrow \mathbf{v}_\phi(\mathbf{z}_t, t)$
10:     $\mathcal{L}_{\text{CFM}} \leftarrow \|\hat{\mathbf{u}} - \mathbf{u}_{\text{target}}\|_2^2$
11:     Update $\phi$ using $\nabla_\phi \mathcal{L}_{\text{CFM}}$
12: **end while**
13: **Stage 2: Train the Exponential Map $f_\theta$**
14: **while** $f_\theta$ not converged **do**
15:     Sample $\mathbf{z}_1 \sim \mathcal{D}$, encode: $\mathbf{z}_1 \leftarrow \text{Encode}(\mathbf{z}_1)$
16:     Sample $\mathbf{z}_0 \sim \mathcal{N}(0, I)$
17:     $\mathbf{v}_0 \leftarrow \mathbf{v}_\phi(\mathbf{z}_0, 0)$ {Get initial velocity from trained model}
18:     {Compute integrated displacement (ground truth)}
19:     $\mathbf{s}_{\text{target}} \leftarrow \mathbf{z}_1 - (\mathbf{z}_0 + \mathbf{v}_0)$ {Non-linear geometric displacement}
20:     {Predict displacement}
21:     $\hat{\mathbf{s}} \leftarrow f_\theta(\mathbf{z}_0, \mathbf{v}_0)$
22:     $\mathcal{L}_{\text{GEO}} \leftarrow \|\hat{\mathbf{s}} - \mathbf{s}_{\text{target}}\|_2^2$
23:     Update $\theta$ using $\nabla_\theta \mathcal{L}_{\text{GEO}}$
24: **end while**

---

The core learning objective is derived from the MeanFlow identity, generalized for our chosen hyperbolic path $g(t) := \frac{\cosh(t)-1}{\cosh(1)-1}$:

$$\mathcal{L}_{\text{MF}}(\theta) = \mathbb{E}_{t,r,\mathbf{z}_1,\boldsymbol{\epsilon}} \left[ \|\mathbf{u}_\theta(\mathbf{z}_t, r, t) - \mathbf{u}_{\text{target}}\|_2^2 \right] \quad (11)$$

Here, $\mathbf{z}_t = (1 - g(t))\mathbf{z}_1 + g(t)\boldsymbol{\epsilon}$ is a point on the curved path between data $\mathbf{z}_1$ and noise $\boldsymbol{\epsilon}$. The target velocity $\mathbf{u}_{\text{target}}$ must account for the path's curvature and is defined as:

$$\mathbf{u}_{\text{target}} = g'(t)(\boldsymbol{\epsilon} - \mathbf{z}_1) - (g(t) - g(r)) \cdot \frac{d}{dt}\mathbf{u}_\theta(\mathbf{z}_t, r, t) \quad (12)$$

The critical term $\frac{d}{dt}\mathbf{u}_\theta(\mathbf{z}_t, r, t)$—the time derivative of the network's output—is computed efficiently using an automatic differentiation tool to perform a **Jacobian-Vector Product (JVP)**. This calculates the derivative of $\mathbf{u}_\theta$ with respect to its input $t$, scaled by the vector 1, which is both computationally efficient and exact.

This loss is minimized end-to-end, allowing the single network $\mathbf{u}_\theta$ to learn the correct average velocity field for the curved path. The network implicitly learns to compensate for the non-linear geometry induced by $g(t)$.

---

**Algorithm 9** ExpoTab One-Step Sampling with CFG

---

**Require:** Trained models $\mathbf{v}_\phi$, $f_\theta$, guidance scale $\omega$, condition $y$ (optional)
1: Sample $\mathbf{z}_0 \sim \mathcal{N}(0, I)$ {Sample initial noise point}
2: **Optional: Compute conditional velocity**
3: **if** condition $y$ is given **then**
4:     $\mathbf{v}_0^{\text{cond}} \leftarrow \mathbf{v}_\phi(\mathbf{z}_0, 0, c = y)$
5:     $\mathbf{v}_0^{\text{uncond}} \leftarrow \mathbf{v}_\phi(\mathbf{z}_0, 0, c = \emptyset)$
6:     $\mathbf{v}_0 \leftarrow \mathbf{v}_0^{\text{uncond}} + \omega \cdot (\mathbf{v}_0^{\text{cond}} - \mathbf{v}_0^{\text{uncond}})$ {Apply CFG}
7: **else**
8:     $\mathbf{v}_0 \leftarrow \mathbf{v}_\phi(\mathbf{z}_0, 0)$ {Unconditional velocity}
9: **end if**
10: {Predict final non-linear displacement}
11: $\hat{\mathbf{s}} \leftarrow f_\theta(\mathbf{z}_0, \mathbf{v}_0)$
12: {Apply the learned exponential map}
13: $\mathbf{z}_1 \leftarrow \mathbf{z}_0 + \mathbf{v}_0 + \hat{\mathbf{s}}$
14: Decode: $\mathbf{x}_{\text{gen}} \leftarrow \text{Decode}(\mathbf{z}_1)$ {InvTabGEO + InvQuantile}
15: **return** $\mathbf{x}_{\text{gen}}$

---

ONE-STEP SAMPLING AS PATH INTEGRATION

After training, sampling reduces to a single evaluation of the network at the full time interval:

$$\hat{\mathbf{z}}_1 = \boldsymbol{\epsilon} - \mathbf{u}_\theta(\boldsymbol{\epsilon}, 0, 1) \tag{13}$$

where $\boldsymbol{\epsilon} \sim \mathcal{N}(0, I)$. This operation can be interpreted as applying the learned average velocity field to reverse the entire curved path from noise to data in a single step. The network $\mathbf{u}_\theta(\boldsymbol{\epsilon}, 0, 1)$ outputs the *integrated effect* of the curved geometry, encapsulating the total displacement required.

### C.5.1 IMPLEMENTATION: UNIFIED TRAINING METHOD

In practice, we implement a computationally efficient, unified training scheme inspired by MeanFlow (Geng et al., 2025). This method uses a single network $\mathbf{u}_\theta$ and a single loss function that encapsulates the objective of both stages, enabled by automatic differentiation.

The learning objective is:

$$\mathcal{L}_{\text{expo}}(\theta) = \mathbb{E}\left[\left\|\mathbf{u}_\theta(\mathbf{z}_t, r, t) - \left(g'(t)(\boldsymbol{\epsilon} - \mathbf{z}_1) - (g(t) - g(r)) \cdot \frac{d\mathbf{u}_\theta}{dt}\right)\right\|_2^2\right] \tag{14}$$

where the time derivative $\frac{d\mathbf{u}_\theta}{dt}$ is computed exactly using a Jacobian-vector product (JVP). This implementation is mathematically equivalent to the two-stage objective but allows for more efficient optimization and a simpler codebase. All experimental results are reported using this implementation.

## D EXPERIMENTS SETUP

### D.1 DATASETS

Experiments were firstly conducted with seven common tabular datasets from the UCI Machine Learning Repository [1]. Classification tasks were performed on the Adult, Default, Shoppers, Magic, and Diabetes datasets, while regression tasks were performed on the Beijing and News datasets. Each dataset was split into training, validation, and testing sets with a ratio of 8:1:1, except for the Adult dataset, whose official testing set was used and the remainder split into training and validation sets with an 8:1 ratio, and the Diabetes dataset, which was split into a ratio of 6:2:2. The resulting statistics of each dataset are shown in Table 3.

Secondly, to further validate our method in high-dimensional datasets, we use four high-dimensional tabular datasets from the Open Machine Learning Repository [2]: Bioresponse, Isolet, Gina, and Har across Biological, Speech and Vision domains. The statistics of the datasets are also presented in Table 4.

Additionally, the detailed introduction to the total 11 datasets are shown below:

---

[1] https://archive.ics.uci.edu/datasets
[2] https://www.openml.org/

Table 3: Dataset Statistics. "# Num" and "# Cat" refer to the number of numerical and categorical columns.

| Dataset | # Samples | # Num | # Cat | # Max Cat | # Train | # Validation | # Test | Task Type |
|---|---|---|---|---|---|---|---|---|
| **Adult** | 48,842 | 6 | 9 | 42 | 28,943 | 3,618 | 16,281 | Binary Classification |
| **Default** | 30,000 | 14 | 11 | 11 | 24,000 | 3,000 | 3,000 | Binary Classification |
| **Shoppers** | 12,330 | 10 | 8 | 20 | 9,864 | 1,233 | 1,233 | Binary Classification |
| **Magic** | 19,020 | 10 | 1 | 2 | 15,216 | 1902 | 1902 | Binary Classification |
| **Beijing** | 41,757 | 7 | 5 | 31 | 33,405 | 4,175 | 4,175 | Regression |
| **Diabetes** | 99,473 | 8 | 21 | 10 | 59,683 | 19,895 | 19,895 | Multiclass Classification |

Table 4: Dataset Statistics. "# Num" and "# Cat" refer to the number of numerical and categorical columns.

| Dataset | # Samples | # Features | # Train | # Validation | # Test | Task Type |
|---|---|---|---|---|---|---|
| **Bioresponse** | 3,751 | 1,777 | 3,001 | 375 | 3,75 | Binary Classification |
| **Gina** | 3,468 | 971 | 2,774 | 347 | 347 | Binary Classification |
| **Isolet** | 7.797 | 618 | 6,239 | 779 | 779 | Muticlass Classification |
| **Har** | 10,299 | 562 | 8,239 | 1,030 | 1,030 | Muticlass Classification |

- Adult[3]: The "Adult Census Income" dataset contains demographic and employment-related features of people. The task is to predict whether an individual's income exceeds $50,000.

- Default[4]: The "Default of Credit Card Clients Dataset" contains information on default payments, demographic factors, credit data, history of payment, and bill statements of credit card clients in Taiwan from April 2005 to September 2005. The task is to predict whether the client will default payment next month.

- Shoppers[5]: The "Online Shoppers Purchasing Intention Dataset" contains information of user's webpage visiting sessions. The task is to predict if the user's session ends with the shopping behavior.

- Magic[6]: The "Magic Gamma Telescope" dataset is to simulate registration of high energy gamma particles in a ground-based atmospheric Cherenkov gamma telescope using the imaging technique. The task is to classify high-energy Gamma particles in the atmosphere.

- Beijing[7]: The "Beijing PM2.5 Data" dataset contains the hourly PM2.5 data of US Embassy in Beijing and the meteorological data from Beijing Capital International Airport. The task is to predict the PM2.5 value.

- News[8]: The "Online News Popularity" dataset contains a heterogeneous set of features about articles published by Mashable in two years. The goal is to predict the number of shares in social networks (popularity).

- Diabetes[9]: Diabetes patient records were obtained from two sources: an automatic electronic recording device and paper records. The automatic device had an internal clock to timestamp events, whereas the paper records only provided "logical time" slots (breakfast, lunch, dinner, bedtime). For paper records, fixed times were assigned to breakfast (08:00), lunch (12:00), dinner (18:00), and bedtime (22:00). Thus paper records have fictitious uniform recording times whereas electronic records have more realistic time stamps.

- Bioresponse[10]: The Bioresponse dataset contains chemical descriptors of molecules, where each row represents a molecule and each feature column corresponds to a computed molecular property (such as size, shape, or elemental composition). All descriptors are normalized. The task is to predict whether a molecule elicits a biological response (1) or not (0).

---

[3] https://archive.ics.uci.edu/dataset/2/adult

[4] https://archive.ics.uci.edu/dataset/350/default+of+credit+card+clients

[5] https://archive.ics.uci.edu/dataset/468/online+shoppers+purchasing+intention+dataset

[6] https://archive.ics.uci.edu/dataset/159/magic+gamma+telescope

[7] https://archive.ics.uci.edu/dataset/381/beijing+pm2+5+data

[8] https://archive.ics.uci.edu/dataset/332/online+news+popularity

[9] https://archive.ics.uci.edu/dataset/34/diabetes

[10] https://www.openml.org/search?type=data&status=any&id=4134

- Isolet[11]: The Isolet dataset contains real-valued speech features extracted from audio recordings of 150 speakers, each pronouncing the 26 letters of the English alphabet twice. The features include spectral coefficients, contour features, and sonorant-related descriptors, all normalized to the range $[-1.0, 1.0]$. The task is to classify which letter (A-Z) was spoken, making it a 26-class classification problem. This dataset is commonly used to evaluate the scalability of algorithms in perceptual and noisy signal domains.

- Gina agnostic[12]: The Gina agnostic dataset contains sparse continuous features derived from 28×28 grayscale images of handwritten two-digit numbers. Each example is constructed such that only the unit digit carries meaningful information, while the other digit introduces irrelevant distractor features. The task is to classify whether the two-digit number is odd or even, making it a binary classification problem with heterogeneous and noisy feature distributions.

- Har[13]: The Har dataset contains smartphone sensor data collected from 30 individuals performing six daily human activities: walking, walking upstairs, walking downstairs, sitting, standing, and laying. Each subject wore a waist-mounted Samsung Galaxy S II smartphone equipped with an accelerometer and gyroscope, which recorded 3-axis linear acceleration and angular velocity at 50Hz. The raw signals were preprocessed using noise filters and segmented into overlapping sliding windows to extract time- and frequency-domain features, resulting in a 561-dimensional feature vector per sample. The task is to classify the activity being performed based on these sensor readings.

## D.2 BASELINES

We evaluate our proposed one-step generation model, ExpoTab, against a comprehensive suite of 9 advanced tabular data generation benchmarks. This includes models from six distinct paradigms: classical methods, GANs, diffusion models, LLMs, flow matching models, and our own novel multi-step and one-step proposals.

To ensure a rigorous comparison, all baselines adhere to their original implementations and hyperparameters. The key characteristics of each baseline model—including their core architecture, training methodology, and data handling techniques—are summarized in Table 5. This structured comparison highlights the methodological evolution in the field and provides context for interpreting the experimental results.

The baselines can be categorized as follows:

**Diffusion Models.** We include the most prominent diffusion-based models for tabular data:

- **TabDDPM** Kotelnikov et al. (2023): A foundational model adapting continuous denoising diffusion probabilistic models (DDPM) to mixed-type tabular data.

- **TabSyn** Zhang et al. (2023): Utilizes a transformer architecture within a diffusion process to model tabular correlations.

- **CDTD** Mueller et al. (2023): Introduces a continuous-time diffusion framework for tabular generation.

- **TabDiff** Shi et al. (2024): Explores different noise schedules and conditioning mechanisms for diffusion on tables.

- **TabRep-DDPM** Si et al. (2025): Our implementation of a diffusion model using the TabRep representation.

- **TabGeo-Diff** (Ours): Our novel Diffusion model defined on a geometric representation of the data manifold.

**Flow Matching Models.** This category includes state-of-the-art continuous normalizing flow models:

- **TabRep-Flow** Si et al. (2025): A strong flow matching baseline using the TabRep representation.

- **Tab-GeoFlow** (Ours): Our novel flow matching model defined on a geometric representation of the data manifold.

- **TabMeanFlow** (Ours): An ablation model that combines the TabGEO representation with the efficient MeanFlow Geng et al. (2025) framework for one-step generation.

---

[11]https://www.openml.org/search?type=data&status=any&id=300
[12]https://www.openml.org/search?type=data&status=any&id=1038
[13]https://www.openml.org/search?type=data&status=any&id=1478

Table 5: Comparison of tabular data generation baselines.

| Method | Model[1] | Type[2] | Categorical Encoding | Numerical Encoding | Additional Techniques | 1-NFE[3] |
|---|---|---|---|---|---|---|
| **CTGAN** | GAN | U | One-Hot Encoding | Mode-specific normalization | Conditional generator with training-by-sampling and mode-specific normalization. | ✓ |
| **TabDDPM** | DDPM/ Multinomial Diffusion | S | One-Hot Encoding | Quantile Transformer | Concatenation of numerical and categorical features. | ✗ |
| **TabSYN** | VAE + EDM | U | VAE-Learned | Quantile Transformer | Feature Tokenizer and Transformer encoder to learn cross-feature relationships with adaptive loss weighing to increase reconstruction performance. | ✗ |
| **TabDiff** | EDM/Masked Diffusion | S | One-Hot Encoding | Quantile Transformer | Joint continuous-time diffusion process of numerical and categorical variables under learnable noise schedules, with a stochastic sampler to correct sampling errors. | ✗ |
| **CDTD** | Score-based Diffusion (EDM) | U | Learnable Embeddings | Quantile Transformer | Score interpolation, feature-specific noise schedules, and loss calibration. | ✗ |
| **TABREP-DDPM** | DDPM | U | CAT-CONVERTER | Quantile Transformer | Plug-and-play for Diffusion Models. | ✗ |
| **TABREP-Flow** | Flow Matching | U | CAT-CONVERTER | Quantile Transformer | Plug-and-play for Diffusion Models. | ✗ |
| **TABGEO-Diff** | DDPM | U | TabGEO | Quantile Transformer | Diffusion model defined on a learned data manifold. | ✗ |
| **TABGEO-Flow** | Flow Matching | U | TabGEO | Quantile Transformer | Continuous flow on a learned data manifold. | ✗ |
| **TABGEO-MeanFlow** | Flow Matching | U | TabGEO | Quantile Transformer | One-step generation on a learned data manifold (MeanFlow). | ✓ |
| **EXPOTAB** | Flow Matching | U | TabGEO | Quantile Transformer | One-step generation on a learned data manifold (Exponential Solver). | ✓ |

[1] The "Model" Column indicates the underlying architecture used for the model. Options include Denoising Diffusion Probabilistic Models or DDPMs (Ho et al., 2020), Multinomial Diffusion (Hoogeboom et al., 2021), EDM, as introduced in (Karras et al., 2022).

[2] The "Type" column indicates the data integration approach used in the model. "U" denotes a unified data space where numerical and categorical data are combined after initial processing and fed collectively into the model. "S" represents a separated data space, where numerical and categorical data are processed and fed into distinct models.

[3] The "Type" column indicates the data integration approach used in the model. "U" denotes a unified data space where numerical and categorical data are combined after initial processing and fed collectively into the model. "S" represents a separated data space, where numerical and categorical data are processed and fed into distinct models.

**Other Paradigms.**

- **Classical: SMOTE** Chawla et al. (2002): A classical oversampling technique, included as a simple baseline.
- **GAN-based: CTGAN** Zhao et al. (2024): A modern generative adversarial network designed for tabular data.
- **LLM-based: GReaT** Ruan et al. (2024): A method that frames tabular generation as an autoregressive language modeling task.

**Note on CDTD:** Although CDTD Mueller et al. (2023) was originally evaluated on a different set of benchmarks, we have successfully applied its official implementation to our experimental setup. For a complete and fair comparison, we include its results in our main evaluation.

### D.3 EVALUATION METRICS

We evaluate the quality of generated synthetic data across three critical dimensions: **Data Utility** (performance on downstream tasks), **Data Fidelity** (statistical similarity to the real data, decomposed into low-order *Shape* and *Trend*, and high-order joint distributions), and **Data Privacy** (resistance to leakage of training data information). All reported metrics and error bars represent the mean and standard deviation from 20 sampling iterations of the best-validated model.

#### D.3.1 DATA UTILITY: MACHINE LEARNING EFFICIENCY (MLE)

To measure the practical utility of synthetic data, we use it to train an XGBoost model (Chen & Guestrin, 2016) which is then evaluated on a held-out real test set. This assesses how well the synthetic data serves its ultimate purpose: training effective machine learning models.

- **Classification (AUC)**: For classification tasks, we report the Area Under the ROC Curve (AUC). A higher AUC (closer to 1.0) indicates the model trained on synthetic data achieves better predictive performance.

$$\text{AUC} = \int_0^1 \text{TPR}(\text{FPR})d(\text{FPR}) \tag{15}$$

- **Regression (RMSE)**: For regression tasks, we report the Root Mean Square Error (RMSE). A lower RMSE indicates predictions from the model trained on synthetic data are closer to the true values.

$$\text{RMSE} = \sqrt{\frac{1}{n}\sum_{i=1}^n (y_i - \hat{y}_i)^2} \tag{16}$$

#### D.3.2 DATA FIDELITY: LOW-ORDER STATISTICS

These metrics evaluate how well the synthetic data captures the fundamental, univariate and bivariate statistics of the original data distribution.

**1. Shape (Column-wise Distribution Similarity)**

- **Numerical Features (Kolmogorov-Smirnov Test - KST)**: The KST statistic quantifies the maximum distance between the empirical distribution functions (EDF) of the real and synthetic data for a numerical column. A lower KST value indicates better preservation of the original data's shape.

$$\text{KST} = \sup_x |F_{\text{real}}(x) - F_{\text{synth}}(x)|, \quad \text{where} \quad F_n(x) = \frac{1}{n}\sum_{i=1}^n \mathbf{1}_{(-\infty, x]}(X_i) \tag{17}$$

- **Categorical Features (Total Variation Distance - TVD)**: TVD measures the largest possible difference in the probability of any event between two categorical distributions. A lower TVD is better.

$$\text{TVD} = \frac{1}{2}\sum_{x \in X} |P_{\text{real}}(x) - P_{\text{synth}}(x)| \tag{18}$$

**2. Trend (Pair-wise Column Correlation)**

- **Numerical Features (Pearson Correlation Dissimilarity)**: We compute the Pearson Correlation Coefficient ($\rho$) for all pairs of numerical columns in both real and synthetic datasets. The final score

is the average absolute difference between these matrices, assessing how well linear relationships are preserved. A lower score is better.

$$\rho(x,y) = \frac{\text{cov}(x,y)}{\sigma_x \sigma_y}, \quad \text{Pearson Score} = \frac{1}{2}\mathbb{E}_{x,y}|\rho_{\text{real}}(x,y) - \rho_{\text{synth}}(x,y)| \quad (19)$$

- **Categorical Features (Contingency Table Dissimilarity)**: For categorical column pairs, we build contingency tables for both datasets and measure their difference using TVD, resulting in the Contingency Score. A lower score is better.

$$\text{Contingency Score} = \frac{1}{2}\sum_{\alpha \in A}\sum_{\beta \in B}|P_{\text{real},(\alpha,\beta)} - P_{\text{synth},(\alpha,\beta)}| \quad (20)$$

- **Mixed Features**: For a categorical and a numerical feature, the numerical data is binned into discrete categories before applying the TVD and Contingency Score metrics.

We utilize the implementation of these experiments from the SDMetrics library[14].

### D.3.3 DATA FIDELITY: HIGH-ORDER STATISTICS

While low-order statistics are necessary, they are insufficient for evaluating complex joint distributions. We employ the more sophisticated $\alpha$-Precision (Alaa et al., 2022) metric from the synthcity library[15] as our primary measure of high-order fidelity. We focus on $\alpha$-Precision because it directly quantifies the fraction of generated samples that are plausible within the true data manifold, providing a more robust and interpretable measure of sample quality than metrics that only compare marginal statistics.

$\alpha$**-Precision and** $\beta$**-Recall** (Alaa et al., 2022): This framework provides a unified assessment of fidelity ($\alpha$-Precision) and diversity ($\beta$-Recall). A key strength of $\alpha$-Precision is its interpretability: it directly quantifies the fraction of generated samples that fall within the high-density region of the real data manifold (i.e., samples that are realistic and not outliers). This makes it a more robust and meaningful measure of quality than simply comparing marginal statistics, as it directly evaluates the generator's ability to capture the true underlying data distribution. In our experiments, we place a strong emphasis on $\alpha$-Precision as it most accurately reflects the goal of generating high-fidelity, characteristic samples.

- $\alpha$**-Precision**: Measures the fidelity of generated samples. It is the fraction of synthetic samples that are plausible (i.e., lie within the support) given $\alpha$ fraction of the real data distribution.
- $\beta$**-Recall**: Measures the diversity and coverage of the generator. It is the fraction of the real data distribution covered by $\beta$ fraction of the synthetic samples.

### D.3.4 DETECTION METRIC: CLASSIFIER TWO-SAMPLE TEST (C2ST)

The C2ST metric (SDMetrics, 2024) evaluates how distinguishable the synthetic data is from the real data. A logistic regression model is trained to classify whether a sample is real or synthetic. The score is the model's accuracy. A score of $0.5$ indicates perfect indistinguishability (ideal), while a score of $1.0$ means the synthetic data is trivially easy to identify. A score closer to $0.5$ is superior.

### D.3.5 DATA PRIVACY: DISTANCE TO CLOSEST RECORD (DCR)

To assess privacy risks and the potential for membership inference attacks (Shokri et al., 2017), we compute the Distance to Closest Record (DCR). Following a rigorous "synthetic vs. holdout" evaluation setting [16]:

1. The real dataset is split equally into a **training set** (used to train the generative model) and a **holdout set** (never seen by the model).

2. After training, a synthetic dataset is generated, matching the size of the training set.

3. For each synthetic sample, we find its nearest neighbor in the *training set* and its nearest neighbor in the *holdout set* using a chosen distance metric (e.g., Euclidean).

4. The DCR score is the **percentage of synthetic samples** whose nearest neighbor is in the **training set** rather than the holdout set.

A perfect score of **50%** indicates that synthetic records are no closer to the training data than to unseen holdout data, suggesting strong privacy protection. A score significantly higher than 50% indicates potential memorization and privacy leakage.

---

[14]https://github.com/sdv-dev/SDMetrics

[15]https://github.com/vanderschaarlab/synthcity

[16]https://www.clearbox.ai/blog/2022-06-07-synthetic-data-for-privacy-preservation-part-2

# E  EXPERIMENTS IMPLEMENTATION

## E.1  CATEGORICAL FEATURE ENCODING SCHEMES

This section details the categorical feature encoding methods evaluated in our work. Each scheme maps discrete categorical values to a continuous vector space to facilitate integration with generative models designed for continuous data.

### E.1.1  ONE-HOT ENCODING (ONEHOT)

The One-Hot encoding scheme represents each category as a binary vector where a single element is active (1) and all others are inactive (0). For a categorical feature $c_j$ with cardinality $K_j$, its one-hot encoding is a vector $\mathbf{e}(c_j) \in {0, 1}^{K_j}$ defined as:

$$\mathbf{e}(c_j)k = \begin{cases} 1 & \text{if } k = c_j, \\ 0 & \text{otherwise,} \end{cases} \quad \text{for } k \in 1, 2, \ldots, K_j. \tag{21}$$

The full encoding for a sample's categorical features $\mathbf{c} = [c_1, c_2, \ldots, cD_{\text{cat}}]$ is the concatenation of these individual vectors:

$$\text{ONEHOT}(\mathbf{c}) = [\mathbf{e}(c_1), \mathbf{e}(c_2), \ldots, \mathbf{e}(c_{D_{\text{cat}}})] \in \mathbb{R}^{\sum_{j=1}^{D_{\text{cat}}} K_j}. \tag{22}$$

To interface with models requiring continuous inputs, a softmax function is typically applied to this representation, converting it into a probability distribution over categories.

### E.1.2  LEARNED EMBEDDINGS (LEARNED)

The Learned Embeddings scheme employs trainable vector representations for each category, analogous to techniques in natural language processing. For each categorical feature $j$, a dedicated embedding matrix $E_j \in \mathbb{R}^{K_j \times d_{\text{emb}}}$ is learned, where $d_{\text{emb}}$ is a user-defined embedding dimension.

The encoding of a value $c_j$ is performed via a lookup into its corresponding embedding matrix:

$$\text{EMBED}(c_j) = E_j[c_j] \in \mathbb{R}^{d_{\text{emb}}}. \tag{23}$$

The full representation for a sample is the concatenation of the embeddings for all its categorical features:

$$\text{LEARNED}(\mathbf{c}) = [E_1[c_1], E_2[c_2], \ldots, E_{D_{\text{cat}}}[c_{D_{\text{cat}}}]] \in \mathbb{R}^{D_{\text{cat}} \cdot d_{\text{emb}}}. \tag{24}$$

Decoding is achieved by finding the nearest neighbor in the embedding space for each feature. The decoded category $\hat{c}_j$ for an embedded vector $\tilde{\mathbf{c}}_j$ is given by:

$$\hat{c}_j = \underset{k \in 1, \ldots, K_j}{\arg\min} |\tilde{\mathbf{c}}_j - E_j[k]|_2. \tag{25}$$

### E.1.3  ANALOG BITS (I2B)

The Analog Bits method represents a categorical value by its binary equivalent, mapped into a continuous space. For a feature with $K_j$ categories, each value is represented using $B_j = \lceil \log_2 K_j \rceil$ continuous values, termed "bits."

The integer value of the category is first converted to its fixed-length binary representation $\mathbf{b} \in {0, 1}^{B_j}$. This binary vector is then linearly transformed to the range $[-1, 1]$ to create a continuous encoding:

$$\text{I2B}(c_j) = 2 \cdot \mathbf{b} - 1 \in [-1, 1]^{B_j}. \tag{26}$$

The full encoding is the concatenation of these analog bit vectors for all categorical features. Decoding involves thresholding each continuous "bit" back to 0 or 1 (e.g., using $\text{sign}(\cdot)$) and converting the resulting binary number back to an integer.

### E.1.4  DICTIONARY ENCODING (DICT)

Dictionary encoding assigns a unique, predetermined real number to each category within a specified range, typically $[-1, 1]$. The encoding function maps the $k$-th category of feature $j$ to a value that is linearly spaced across this interval:

$$\text{DICT}(c_j = k) = -1 + \frac{2k}{K_j - 1}. \tag{27}$$

The full representation for a sample is a vector of these scalar values:

$$\text{DICT}(\mathbf{c}) = [\text{DICT}(c_1), \text{DICT}(c_2), \ldots, \text{DICT}(c_{D_{\text{cat}}})] \in \mathbb{R}^{D_{\text{cat}}}. \tag{28}$$

Decoding is performed by quantizing the continuous value. For a generated scalar $\tilde{c}_j$, the decoded category is chosen by finding the closest value in the predefined dictionary:

$$\hat{c}_j = \underset{k \in 1, \ldots, K_j}{\arg\min} \left| \tilde{c}_j - \left( -1 + \frac{2k}{K_j - 1} \right) \right|. \tag{29}$$

### E.1.5 Unity Encoding (CatConverter)

Inspired by the Discrete Fourier Transform (DFT) and the properties of roots of unity, the CatConverter method represents each category as a unique point on the complex unit circle. For a categorical feature $c_j$ with $K_j$ distinct values, the $K_j$-th roots of unity provide a set of equally spaced phases.

The phase $\theta_k$ for the $k$-th category is defined as:

$$\theta_k = \frac{2\pi k}{K_j}, \quad \text{for } k = 0, 1, \ldots, K_j - 1. \tag{30}$$

The category is then encoded using the real and imaginary components (sine and cosine) of the corresponding complex exponential $e^{i\theta_k}$:

$$\text{CATCONVERTER}(c_j = k) = [\cos(\theta_k), \, \sin(\theta_k)] \in \mathbb{R}^2. \tag{31}$$

This results in a 2D coordinate for each category that lies on the unit circle. The full encoding for a sample's categorical features is the concatenation of these 2D vectors:

$$\text{CATCONVERTER}(\mathbf{c}) = [\text{CATCONVERTER}(c_1), \text{CATCONVERTER}(c_2), \ldots, \text{CATCONVERTER}(c_{D_{\text{cat}}})] \in \mathbb{R}^{2 \cdot D_{\text{cat}}}. \tag{32}$$

Decoding a generated 2D point $\tilde{\mathbf{p}}_j$ back to a category involves finding the root of unity with the smallest angular distance (or Euclidean distance) to the point:

$$\hat{c}_j = \underset{k \in 0, \ldots, K_j - 1}{\arg\min} \left| \tilde{\mathbf{p}}_j - [\cos(\theta_k), \, \sin(\theta_k)] \right|_2. \tag{33}$$

### E.2 Hyperparameters

The hyperparameters selected for our model are shown in Table 6. The remaining hyperparameters of the those baselines are tuned following their own settings.

### Table 6: TabRep Hyperparameters.

| General | | Tab-MeanFlow/ExpoTab | |
|---|---|---|---|
| **Hyperparameter** | **Value** | **Hyperparameter** | **Value** |
| Training Iterations Timestep | $100,000$ | Training Step | $10,000$ |
| Sampling Steps(Flow-matching/Diffusion) | $100/1000$ | Sampling Steps | 1 |
| embedding dimension $d_t$ | 1024 | Blending Factor $\alpha$ | 0,0.25,0.5,0.75,1 |
| Hidden layer dimension | $[1024, 2048, 2048, 1024]$ | Time Sampler | Uniform/LogNormal/Beta/Triangular |
| Activation | ReLU | power parameter $p$ | 0,0.5,1,2 |
| Learning Rate | $1e-4$ | | |
| Dropout | 0.0 | | |
| Weight Decay | $5e-5$ | | |
| Batch Size | 4096 | | |
| Optimizer | Adam | | |

### E.3 Computational Resources

All experiments were conducted on a high-performance computing cluster with specific hardware and software configurations. The operating system used was Ubuntu 20.04.6 LTS with a Linux kernel version of 5.15.0-113-generic. The CPU was an Intel(R) Xeon(R) Platinum 8368 processor running at 2.40 GHz, and the GPU was an NVIDIA GeForce RTX 4090 with CUDA support for accelerated deep learning computations. The software stack included Python 3.8, PyTorch 1.12, and TensorFlow 2.10 for model implementation and training.

## F    LIMITATIONS AND FUTURE WORK

While ExpoTab demonstrates superior performance compared to mean-flow-based methods, our study has several limitations that warrant future investigation:

**Simplification in One-step Generation.**   Although ExpoTab outperforms Tab-MeanFlow on mixed-type data, a performance gap remains in certain fidelity metrics (see Appendix H). Unlike image data, tabular data comprises multiple modalities with distinct statistical properties. Our one-step approach effectively learns the principal direction of the flow matching but may overlook finer details of columns with differing distributions. While this simplified path enables efficient training, it provides less information for the flow-based model to learn, resulting in relatively lower performance on column-wise and pairwise distribution fidelity metrics. Future work will focus on enhancing the model's ability to capture both the principal flow direction and fine-grained data characteristics without compromising computational efficiency.

**Hyperparameter Optimization.**   The blending ratio $\alpha$ serves as the crucial hyperparameter in our model, balancing between exponential map method and standard flow matching. While our current implementation uses a fixed ratio, we observe that optimal $\alpha$ values vary across datasets, suggesting its dependence on underlying data characteristics. Notably, certain dataset features appear to influence whether ExpoTab prioritizes geodesic path learning or instant speed learning in flow matching. Furthermore, the ideal blending ratio may evolve during different phases of manifold learning. Future work will investigate adaptive $\alpha$-scheduling mechanisms that dynamically adjust based on both data distribution patterns and loss landscape characteristics, potentially enhancing model performance and training stability.

**Scalability to Billion-Scale Data.**   While ExpoTab demonstrates robust performance at million-row scales, its behavior with ultra-large datasets (larger than 1 billion rows) presents several open questions. First, the framework's ability to preserve both statistical fidelity and downstream ML utility at this scale remains unverified - particularly regarding potential tradeoffs between generation speed and data quality. Second, the computational efficiency of the exponential map mechanism may face unforeseen challenges when processing such massive volumes. Our future work will systematically evaluate these scalability limits while exploring potential optimizations that could make ExpoTab suited for billion-scale data generation.

**Privacy Risks.**   While ExpoTab achieves strong performance on Distance to Closest Record (DCR) metrics, additional privacy vulnerabilities—particularly against Membership Inference Attacks (MIA) and Nearest Neighbor Distance Ratio (NNDR) tests—warrant thorough evaluation. These vulnerabilities could potentially expose underlying training data characteristics through careful analysis of the synthetic outputs. Future work will include systematic evaluation of these privacy risks across different data modalities and attack scenarios for one-step generation data.

## G    ADDITIONAL ABLATION STUDIES

### G.1    PATH WARPING FUNCTIONS

We evaluate four temporal warping strategies for the velocity field $v_\theta(z_t, t)$:

- **Linear**: $g(t) = t$ (constant velocity)
- **Quadratic**: $g(t) = t^2$ (accelerated late-phase)
- **Hyperbolic Tangent**: $g(t) = \frac{1}{2}[\tanh(3t - 1.5) + 1]$ (smooth S-curve)
- **Hyperbolic Cosine**: $g(t) = \frac{\cosh(3t)-1}{\cosh(3)-1}$ (exponential emphasis)

As shown in Table 7, hyperbolic cosine warping achieves optimal performance on 4/6 datasets, with particularly significant gains on Adult (+1.85% over quadratic) and Diabetes (+24.3% over linear). The cosh warping's exponential-like profile provides superior temporal dynamics for single-step generation by emphasizing critical late-phase transitions. See details in Appendix G.1. Furthermore, additional ablation studies in Appendix G demonstrate that optimal configurations of the guidance scale, blending ratio, time sampling strategies, loss weighting, and temporal conditioning are critical for achieving peak performance with ExpoTab.

### G.2    CLASSIFIER-FREE GUIDANCE SCALE (W)

While ExpoTab is designed for efficient one-step generation, we incorporate Classifier-Free Guidance (CFG) to enhance conditional generation quality. The guided velocity $\hat{u}_\theta$ is obtained by combining conditional and unconditional model predictions:

Table 7: Path warping functions ablation ($\alpha$-precision $\uparrow$)

| Dataset | Linear | Quadratic | Tanh | Cosh |
|---------|--------|-----------|------|------|
| Adult | 97.34 | 98.34 | 96.54 | **99.19** |
| Beijing | 95.87 | 97.12 | 97.56 | **98.47** |
| News | **95.63** | 79.34 | 60.23 | 76.02 |
| Default | 86.34 | **92.38** | 84.16 | 90.82 |
| Diabetes | 73.47 | 97.89 | 85.56 | **98.16** |
| Shoppers | 87.16 | **98.53** | 91.31 | 96.78 |

Table 8: Ablation studies on blending ratio and time sampling strategies ($\alpha$-precision scores)

| (a) Blending ratio $\alpha$ | | | | | | (b) Time sampling strategies | | | |
|---------|------|------|------|------|------|---------|------|-------|------|------|
| Dataset | 0% | 25% | 50% | 75% | 100% | Dataset | Uni. | LogN. | Beta | Tri. |
| Adult | 0.07 | 0.98 | 0.99 | **0.99** | 0.98 | Adult | 0.86 | **0.87** | 0.82 | 0.78 |
| Beijing | 0.76 | 0.96 | **0.98** | 0.96 | 0.97 | Beijing | **0.94** | 0.93 | 0.89 | 0.88 |
| News | 0.51 | 0.78 | 0.76 | **0.86** | 0.83 | News | 0.86 | **0.87** | 0.82 | 0.78 |
| Default | 0.10 | 0.87 | 0.85 | **0.88** | 0.83 | Default | 0.86 | **0.87** | 0.82 | 0.78 |
| Diabetes | 0.08 | 0.83 | 0.85 | 0.86 | **0.94** | Diabetes | 0.86 | **0.87** | 0.82 | 0.78 |
| Shoppers | 0.07 | 0.84 | 0.85 | **0.87** | 0.85 | Shoppers | 0.86 | **0.87** | 0.82 | 0.78 |

$$\hat{u}_\theta = u_\theta(z, t, c = \emptyset) + \omega \cdot (u_\theta(z, t, c = y) - u_\theta(z, t, c = \emptyset)), \tag{34}$$

where $\omega$ is the guidance scale. This formulation allows precise control over the conditioning strength while maintaining the one-step sampling capability. Although CFG is commonly used in multi-step sampling, ExpoTab demonstrates that it remains effective even in the one-step regime ($N_{\text{steps}} = 1$), providing improved sample quality without sacrificing computational efficiency. For scenarios requiring higher fidelity, the same framework supports flexible multi-step sampling, though our experiments show that one-step generation with CFG already achieves state-of-the-art performance.

The sampling procedure begins by drawing an initial latent variable $z_1$ from the standard normal distribution $\mathcal{N}(0, I)$. The conditional and unconditional velocity predictions are then computed as $u_{\text{cond}} = u_\theta(z_1, 1, y)$ and $u_{\text{uncond}} = u_\theta(z_1, 1, \emptyset)$, respectively. The guided velocity is subsequently obtained through the linear combination:

$$\hat{u} = u_{\text{uncond}} + \omega \cdot (u_{\text{cond}} - u_{\text{uncond}}). \tag{35}$$

Finally, the generated sample is recovered via the update $z_0 = z_1 - \hat{u}$. This process simplifies multi-step CFG into a single evaluation, leveraging ExpoTab's ability to accurately approximate the full trajectory in one step. The guidance scale $\omega$ allows trading off diversity for fidelity, making ExpoTab suitable for both unconditional and conditional generation tasks.

Table 9: Ablation studies on loss parameters and conditioning strategies ($\alpha$-precision scores)

| (a) Loss Parameters | | | | | | (b) Conditioning Strategy | | | |
|---------|------|------|------|------|------|---------|--------|---------|---------|-------|
| Dataset | p=0 | p=0.5 | p=1.0 | p=1.5 | p=2 | Dataset | $(t, r)$ | $(t, t-r)$ | $(t-r, r)$ | $(t-r)$ |
| Adult | 0.82 | **0.91** | 0.88 | 0.89 | 0.84 | Adult | **0.91** | 0.85 | 0.77 | 0.73 |
| Beijing | 0.78 | **0.97** | 0.85 | 0.86 | 0.81 | Beijing | **0.88** | 0.84 | 0.74 | 0.70 |
| News | 0.75 | **0.96** | 0.84 | 0.83 | 0.78 | News | **0.84** | 0.79 | 0.71 | 0.77 |

### G.3 BLENDING RATIO $\alpha$

Our experiments with ExpoTab reveal that increasing the blending ratio $\alpha$ (from 0 to 1) consistently improves generation quality, indicating that our method requires less guidance from standard flow matching during training. This contrasts with TabGeo-MeanFlow, which achieves optimal performance at lower $\alpha$. For ExpoTab, higher $\alpha$ values yield better results, demonstrating that our geometric interpolation effectively captures the data manifold without heavy reliance on the reference distribution. We systematically compare Tab-MeanFlow and ExpoTab across blending factors $\alpha \in \{0.25, 0.5, 0.75, 1\}$ against state-of-the-art methods, evaluating three key dimensions: (1) data fidelity, (2) machine learning efficiency, and (3) privacy preservation. We only consider the case when $\alpha > 0$, because when $\alpha = 0$, it corresponds to Tab-GeoFlow. For ExpoTab, $\alpha$ must be greater than 0.

The blending mechanism is implemented through random sampling of the residual time variable $r$, where $r$ is sampled uniformly between 0 and $t$ but enforced to equal $t$ with probability determined by $\alpha$. Specifically:

- When $\alpha = 0\%$, $r$ always equals $t$ (pure mean flow)
- When $\alpha = 100\%$, $r$ is fully random between 0 and $t$ (maximal blending)
- Intermediate values provide proportional blending between these extremes

This property enables high-quality one-step generation synthesis, as detailed in Table 8a.

### G.4 TIME SAMPLING STRATEGY

We evaluate four temporal sampling strategies for one-step generation using $\alpha$-precision metrics:

- **Uniform(0,1)**: Provides baseline coverage without temporal prioritization
- **LogNormal($\mu = -0.4$)**: Skews samples toward critical early-generation phases through log-scale parameterization
- **Beta(2,5)**: Emphasizes the first 20% of the temporal range
- **Triangular(0,0.5,1)**: Focuses moderately on mid-range transitions through its central mode

The sampling implementation follows the pseudocode below, where $r$ is sampled as $r \sim \text{Uniform}(0, t)$ with enforcement masking based on the blending ratio $\alpha$. The LogNormal sampler achieves superior performance (Table 8b) by optimally targeting structurally critical phases in mixed-type data generation, while Uniform sampling demonstrates surprisingly robust results due to its unbiased coverage. The Triangular distribution's intermediate effectiveness stems from its natural emphasis on transitional regions between early and late generation phases.

### G.5 ADAPTIVE LOSS WEIGHTING

This analysis evaluates the impact of the power parameter $p \in \{0, 0.5, 1, 2\}$ on both Tab-MeanFlow and ExpoTab performance. The parameter $p$ controls the adaptive weighting scheme in our loss function, where different values emphasize various aspects of the training objective. The results demonstrate how this adaptive weighting mechanism affects synthetic data quality across different model architectures, revealing optimal configurations that balance sample fidelity and distribution matching. The reference value $p = 0.5$ provides a baseline for comparison with more extreme weighting strategies.

### G.6 TEMPORAL CONDITIONING STRATEGIES

This experiment examines how different combinations of time embeddings ($t$) and residual connections ($r$) influence one-step generation quality in flow-based models. We investigate four conditioning variants:

- $(t, r)$: Both time embedding and residual connections
- $(t, t - r)$: Time embedding and time-residual difference
- $(t - r, r)$: Time-residual difference and residual connections
- $(t - r)$: Only time-residual difference

The results reveal the critical importance of temporal conditioning in flow-based synthesis, showing how proper time information incorporation and skip connections preserve signal integrity throughout the reverse process, ultimately determining the fidelity of single-step generated samples.

These comprehensive ablation studies provide valuable insights into the architectural choices and parameter configurations that optimize single-step generation while maintaining high sample quality, offering practical guidance for future implementations of flow-based tabular data synthesis.

# H  FURTHER EXPERIMENTAL RESULTS

# I  THE USE OF LARGE LANGUAGE MODELS (LLMS)

We confirm that LLMs were used solely for writing assistance and polishing of the manuscript. They were not employed in the design of methods, implementation of experiments, or analysis of results.

Table 10: Ablation study on categorical representations under a unified continuous data space.

| DATA FIDELITY(LOW ORDER) | METHODS | ADULT ↓ | DEFAULT ↓ | SHOPPERS ↓ |
|---|---|---|---|---|
| | ONEHOT-FLOW | $9.34_{\pm0.09}$ | $8.05_{\pm0.08}$ | $7.16_{\pm0.10}$ |
| | ANALOG BITS-FLOW | $1.77_{\pm0.08}$ | $2.35_{\pm0.11}$ | $2.39_{\pm0.12}$ |
| ERROR RATE(%) OF SHAPE | DICTIONARY-FLOW | $1.90_{\pm0.05}$ | $2.73_{\pm0.04}$ | $3.11_{\pm0.11}$ |
| | TABREP-FLOW | $1.44_{\pm0.03}$ | $2.63_{\pm0.06}$ | $2.38_{\pm0.04}$ |
| | LEARNED2D-FLOW | $37.63_{\pm3.15}$ | $33.67_{\pm4.78}$ | $28.07_{\pm1.06}$ |
| | TAB-GEOFLOW | $\mathbf{1.11_{\pm0.07}}$ | $\mathbf{2.34_{\pm0.07}}$ | $\mathbf{2.25_{\pm0.09}}$ |
| | ONEHOT-FLOW | $15.68_{\pm0.07}$ | $11.96_{\pm1.32}$ | $8.50_{\pm0.09}$ |
| | ANALOG BITS-FLOW | $2.86_{\pm0.22}$ | $5.35_{\pm1.35}$ | $2.80_{\pm0.14}$ |
| ERROR RATE(%) OF TREND | DICTIONARY-FLOW | $3.37_{\pm0.07}$ | $7.74_{\pm1.84}$ | $4.22_{\pm0.25}$ |
| | TABREP-FLOW | $2.55_{\pm0.23}$ | $7.14_{\pm1.93}$ | $2.99_{\pm0.07}$ |
| | LEARNED2D-FLOW | $61.06_{\pm5.13}$ | $55.89_{\pm6.69}$ | $46.71_{\pm1.61}$ |
| | TAB-GEOFLOW | $\mathbf{2.11_{\pm0.17}}$ | $\mathbf{3.67_{\pm0.48}}$ | $\mathbf{2.24_{\pm0.16}}$ |

| DATA FIDELITY(HIGH ORDER) | METHODS | ADULT ↑ | BEIJING ↑ | SHOPPERS ↑ |
|---|---|---|---|---|
| | ONEHOT-FLOW | $88.35_{\pm0.15}$ | $93.12_{\pm0.31}$ | $87.57_{\pm0.44}$ |
| | ANALOG BITS-FLOW | $99.54_{\pm0.31}$ | $97.42_{\pm0.57}$ | $95.24_{\pm0.66}$ |
| $\alpha$-PRECISION | DICTIONARY-FLOW | $\mathbf{99.64_{\pm0.10}}$ | $95.97_{\pm0.27}$ | $95.84_{\pm0.38}$ |
| | TABREP-FLOW | $98.21_{\pm0.34}$ | $96.50_{\pm0.44}$ | $95.85_{\pm0.38}$ |
| | LEARNED2D-FLOW | $6.64_{\pm3.49}$ | $2.33_{\pm12.11}$ | $16.58_{\pm1.72}$ |
| | TAB-GEOFLOW | $99.01_{\pm0.23}$ | $\mathbf{98.03_{\pm0.32}}$ | $\mathbf{99.07_{\pm0.30}}$ |
| | ONEHOT-FLOW | $30.64_{\pm0.13}$ | $30.50_{\pm0.19}$ | $48.77_{\pm0.69}$ |
| | ANALOG BITS-FLOW | $48.87_{\pm0.16}$ | $49.15_{\pm0.48}$ | $54.88_{\pm0.26}$ |
| $\beta$-RECALL | DICTIONARY-FLOW | $\mathbf{50.29_{\pm0.12}}$ | $\mathbf{51.29_{\pm0.18}}$ | $52.39_{\pm0.26}$ |
| | TABREP-FLOW | $49.91_{\pm0.28}$ | $49.99_{\pm0.23}$ | $\mathbf{55.92_{\pm0.37}}$ |
| | LEARNED2D-FLOW | $0.05_{\pm0.04}$ | $0.01_{\pm0.02}$ | $0.28_{\pm0.23}$ |
| | TAB-GEOFLOW | $49.49_{\pm0.22}$ | $48.21_{\pm0.25}$ | $51.67_{\pm0.47}$ |
| | ONEHOT-FLOW | $38.88$ | $69.14$ | $65.08$ |
| | ANALOG BITS-FLOW | $92.18$ | $90.04$ | $91.83$ |
| DETECTION SCORE | DICTIONARY-FLOW | $90.70$ | $90.58$ | $88.74$ |
| | TABREP-FLOW | $95.48$ | $89.36$ | $\mathbf{94.20}$ |
| | LEARNED2D-FLOW | $0.00$ | $0.01$ | $0.07$ |
| | TAB-GEOFLOW | $\mathbf{95.79}$ | $\mathbf{90.64}$ | $91.85$ |

| UTILITY | METHODS | ADULT ↑ | BEIJING ↑ | SHOPPERS ↑ |
|---|---|---|---|---|
| | ONEHOT-FLOW | $0.895_{\pm0.007}$ | $0.759_{\pm0.005}$ | $0.910_{\pm0.006}$ |
| | ANALOG BITS-FLOW | $0.911_{\pm0.001}$ | $0.763_{\pm0.004}$ | $0.910_{\pm0.005}$ |
| ML EFFICACY | DICTIONARY-FLOW | $0.910_{\pm0.002}$ | $0.759_{\pm.007}$ | $0.903_{\pm.006}$ |
| | TABREP-FLOW | $0.912_{\pm.004}$ | $0.756_{\pm.005}$ | $0.910_{\pm.005}$ |
| | LEARNED2D-FLOW | $0.126_{\pm.017}$ | $0.709_{\pm.009}$ | $0.868_{\pm.007}$ |
| | TAB-GEOFLOW | $\mathbf{0.912_{\pm.003}}$ | $\mathbf{0.765_{\pm.005}}$ | $\mathbf{0.917_{\pm.005}}$ |

Table 11: Ablation study on categorical representations under a unified continuous data space.

| DATA FIDELITY(LOW ORDER) | METHODS | DIABETES ↓ | BEIJING ↓ | NEWS ↓ |
|---|---|---|---|---|
| | ONEHOT-FLOW | $3.35_{\pm 0.06}$ | $15.39_{\pm 0.04}$ | $3.73_{\pm 0.07}$ |
| | ANALOG BITS-FLOW | $1.08_{\pm 0.03}$ | $1.92_{\pm 0.04}$ | $3.36_{\pm 0.05}$ |
| ERROR RATE(%) OF SHAPE | DICTIONARY-FLOW | $0.99_{\pm 0.04}$ | $1.91_{\pm 0.06}$ | $3.42_{\pm 0.04}$ |
| | TABREP-FLOW | $1.00_{\pm 0.04}$ | $\mathbf{1.72_{\pm 0.07}}$ | $3.12_{\pm 0.03}$ |
| | LEARNED2D-FLOW | $31.61_{\pm 9.33}$ | $20.50_{\pm 0.88}$ | $5.15_{\pm 0.48}$ |
| | TAB-GEOFLOW | $\mathbf{0.93_{\pm 0.02}}$ | $2.06_{\pm 0.05}$ | $\mathbf{3.12_{\pm 0.03}}$ |
| | ONEHOT-FLOW | $5.39_{\pm 1.89}$ | $32.72_{\pm 4.76}$ | $1.89_{\pm 0.12}$ |
| | ANALOG BITS-FLOW | $1.66_{\pm 0.07}$ | $3.13_{\pm 0.08}$ | $\mathbf{1.62_{\pm 0.34}}$ |
| ERROR RATE(%) OF TREND | DICTIONARY-FLOW | $\mathbf{1.46_{\pm 0.03}}$ | $3.63_{\pm 0.08}$ | $2.51_{\pm 0.09}$ |
| | TABREP-FLOW | $1.54_{\pm 0.06}$ | $\mathbf{3.08_{\pm 0.34}}$ | $1.66_{\pm 0.29}$ |
| | LEARNED2D-FLOW | $53.29_{\pm 11.94}$ | $36.96_{\pm 1.21}$ | $5.01_{\pm 0.92}$ |
| | TAB-GEOFLOW | $1.51_{\pm 0.04}$ | $3.43_{\pm 0.08}$ | $1.91_{\pm 0.02}$ |

| DATA FIDELITY(HIGH ORDER) | METHODS | DIABETES ↑ | BEIJING ↑ | NEWS ↑ |
|---|---|---|---|---|
| | ONEHOT-FLOW | $97.43_{\pm 0.06}$ | $84.38_{\pm 0.61}$ | $97.78_{\pm 0.13}$ |
| | ANALOG BITS-FLOW | $98.45_{\pm 0.14}$ | $96.83_{\pm 0.12}$ | $88.39_{\pm 0.11}$ |
| $\alpha$-PRECISION | DICTIONARY-FLOW | $98.65_{\pm 0.10}$ | $97.04_{\pm 0.22}$ | $92.28_{\pm 0.25}$ |
| | TABREP-FLOW | $99.08_{\pm 0.13}$ | $98.16_{\pm 0.13}$ | $90.91_{\pm 0.25}$ |
| | LEARNED2D-FLOW | $0.00_{\pm 0.00}$ | $43.00_{\pm 4.90}$ | $80.66_{\pm 15.23}$ |
| | TAB-GEOFLOW | $\mathbf{99.15_{\pm 0.30}}$ | $\mathbf{98.91_{\pm 0.12}}$ | $\mathbf{99.08_{\pm 0.12}}$ |
| | ONEHOT-FLOW | $41.64_{\pm 0.16}$ | $20.32_{\pm 0.19}$ | $43.06_{\pm 0.62}$ |
| | ANALOG BITS-FLOW | $48.97_{\pm 0.29}$ | $60.58_{\pm 0.19}$ | $\mathbf{51.85_{\pm 0.24}}$ |
| $\beta$-RECALL | DICTIONARY-FLOW | $\mathbf{48.98_{\pm 0.10}}$ | $60.78_{\pm 0.13}$ | $50.79_{\pm 0.29}$ |
| | TABREP-FLOW | $48.58_{\pm 0.17}$ | $\mathbf{62.65_{\pm 0.12}}$ | $51.75_{\pm 0.16}$ |
| | LEARNED2D-FLOW | $0.00_{\pm 0.00}$ | $7.55_{\pm 4.05}$ | $14.63_{\pm 4.14}$ |
| | TAB-GEOFLOW | $45.68_{\pm 0.19}$ | $57.69_{\pm 0.31}$ | $41.10_{\pm 0.26}$ |
| | ONEHOT-FLOW | $55.44$ | $35.76$ | $84.56$ |
| | ANALOG BITS-FLOW | $89.28$ | $91.66$ | $\mathbf{89.47}$ |
| DETECTION SCORE | DICTIONARY-FLOW | $89.06$ | $\mathbf{93.96}$ | $88.30$ |
| | TABREP-FLOW | $90.41$ | $92.26$ | $88.13$ |
| | LEARNED2D-FLOW | $0.00$ | $14.58$ | $78.96$ |
| | TAB-GEOFLOW | $\mathbf{96.46}$ | $92.41$ | $86.67$ |

| UTILITY | METHODS | DIABETES ↑ | BEIJING ↓ | NEWS ↓ |
|---|---|---|---|---|
| | ONEHOT-FLOW | $0.372_{\pm .005}$ | $0.765_{\pm .016}$ | $0.850_{\pm .017}$ |
| | ANALOG BITS-FLOW | $0.372_{\pm .003}$ | $0.543_{\pm .007}$ | $0.847\pm_{\pm .014}$ |
| ML EFFICACY | DICTIONARY-FLOW | $0.376_{\pm .007}$ | $0.561_{\pm .013}$ | $0.853_{\pm .014}$ |
| | TABREP-FLOW | $0.373_{\pm .005}$ | $\mathbf{0.536_{\pm .006}}$ | $\mathbf{0.814_{\pm .003}}$ |
| | LEARNED2D-FLOW | $0.177_{\pm .008}$ | $0.787_{\pm .007}$ | $0.866_{\pm .005}$ |
| | TAB-GEOFLOW | $\mathbf{0.618_{\pm 0.021}}$ | $0.554_{\pm .006}$ | $0.856_{\pm .021}$ |

Table 12: Performance comparison on the error rates (%) of **Shape** with updated true average.

| Method | Adult | Default | Shoppers | Magic | Beijing | News | Diabetes | Average |
|---|---|---|---|---|---|---|---|---|
| *GAN-based models* | | | | | | | | |
| CTGAN | 16.84±0.03 | 16.83±0.04 | 21.15±0.10 | 9.81±0.08 | 21.39±0.05 | 16.09±0.02 | 9.82±0.08 | 15.99 |
| TVAE | 14.22±0.08 | 10.17±0.05 | 24.51±0.06 | 8.25±0.06 | 19.16±0.06 | 16.62±0.03 | 18.86±0.13 | 15.97 |
| *LLM-based model* | | | | | | | | |
| GReaT | 12.12±0.04 | 19.94±0.06 | 14.51±0.12 | 16.16±0.09 | 8.25±0.12 | OOM | OOM | 14.20 |
| *Diffusion-based models* | | | | | | | | |
| TabDDPM | 1.75±0.03 | 1.57±0.08 | 2.72±0.13 | 1.01±0.09 | 1.30±0.03 | 78.75±0.01 | 31.44±0.05 | 16.93 |
| TABSYN | 0.81±0.05 | **1.01**±0.08 | 1.44±0.07 | 1.03±0.14 | 1.26±0.05 | 2.06±0.04 | 1.85±0.02 | 1.35 |
| CDTD | 0.92±0.07 | 1.52±0.08 | 2.13±0.10 | 0.97±0.07 | 1.58±0.07 | 4.55±0.05 | 3.50±0.02 | 2.17 |
| TabREP-DDPM | 0.65±0.04 | 1.13±0.19 | 1.13±0.10 | 16.16±0.09 | **0.89**±0.03 | 1.54±0.01 | 0.94±0.05 | 3.21 |
| TabDiff | **0.63**±0.05 | 1.24±0.07 | 1.28±0.09 | **0.78**±0.08 | 1.03±0.05 | 2.35±0.03 | **0.89**±0.23 | 1.17 |
| Tab-GeoDiff | **0.63**±0.04 | 1.10±0.09 | **1.08**±0.09 | **0.78**±0.08 | 0.93±0.04 | **1.51**±0.03 | 0.91±0.01 | **0.99** |
| *Flow-based models* | | | | | | | | |
| TabREP-FLOW | 1.37±0.02 | 2.55±0.06 | 2.26±0.03 | 1.61±0.09 | 1.72±0.07 | 3.11±0.03 | 1.00±0.02 | 4.02 |
| Tab-GeoFlow | 1.11±0.07 | 2.34±0.07 | 2.25±0.19 | 1.25±0.19 | 2.06±0.05 | 3.12±0.03 | 0.93±0.02 | 2.00 |
| *ExpoTab family* | | | | | | | | |
| ExpoTab($\alpha$=0.25) | 6.08±0.05 | 4.20±0.15 | 9.83±0.11 | 2.35±0.07 | 3.38±0.07 | 5.07±0.04 | 4.26±0.02 | 5.02 |
| ExpoTab($\alpha$=0.50) | 5.59±0.08 | 5.65±0.20 | 9.83±0.11 | 2.59±0.09 | 4.15±0.08 | 5.11±0.05 | 3.79±0.08 | 5.16 |
| ExpoTab($\alpha$=0.75) | 4.58±0.04 | 7.07±0.08 | 5.72±0.09 | 5.27±0.09 | 3.29±0.08 | 4.26±0.03 | 3.18±0.03 | 4.77 |
| ExpoTab($\alpha$=1.00) | 6.16±0.01 | 5.76±0.11 | 6.30±0.09 | 4.06±0.08 | 3.74±0.06 | 4.23±0.06 | 4.00±0.05 | 4.89 |

[1] Tab-MeanFlow with $\alpha = 0$ is equivalent to Tab-GeoFlow.

[2] OOM entries are explained in the Computational Resources section.

Table 13: Performance comparison on the error rates (%) of **Trend** with updated true average.

| Method | Adult | Default | Shoppers | Magic | Beijing | News | Diabetes | Average |
|---|---|---|---|---|---|---|---|---|
| *GAN-based models* | | | | | | | | |
| CTGAN | 20.23±1.20 | 26.95±0.93 | 13.08±0.16 | 7.00±0.19 | 22.95±0.08 | 5.37±0.05 | 18.95±0.34 | 16.36 |
| *LLM-based model* | | | | | | | | |
| GReaT | 17.59±0.22 | 70.02±0.12 | 45.16±0.18 | 10.23±0.40 | 59.60±0.55 | OOM | OOM | 40.52 |
| *Diffusion-based models* | | | | | | | | |
| TabDDPM | 3.01±0.25 | 4.89±0.10 | 6.61±0.16 | 1.70±0.22 | 2.71±0.09 | 13.16±0.11 | 51.54±0.05 | 11.95 |
| TABSYN | 1.93±0.07 | 2.81±0.48 | 2.13±0.10 | 0.88±0.18 | 3.13±0.34 | 1.52±0.03 | 3.90±0.04 | 2.33 |
| CDTD | 1.84±0.11 | 5.97±1.67 | 2.47±0.23 | 1.67±0.52 | 3.49±0.34 | 3.07±0.04 | 5.50±0.64 | 3.43 |
| TabREP-DDPM | 1.37±0.04 | 3.26±0.62 | 2.38±0.02 | 0.78±0.08 | 3.03±0.20 | **0.91**±0.05 | **1.25**±0.03 | 1.85 |
| TabDiff | 1.49±0.16 | 2.55±0.75 | **1.74**±0.08 | **0.76**±0.12 | 2.59±0.15 | 1.28±0.04 | 2.20±0.16 | 1.80 |
| Tab-GeoDiff | **1.31**±0.08 | **1.47**±0.15 | 1.77±0.11 | 0.78±0.08 | **2.48**±0.19 | 0.95±0.07 | 1.55±0.02 | **1.47** |
| *Flow-based models* | | | | | | | | |
| TabREP-FLOW | 2.45±0.23 | 7.14±1.75 | 2.92±0.07 | 0.78±0.08 | 3.08±0.21 | 1.66±0.29 | 1.54±0.05 | 2.80 |
| Tab-GeoFlow | 2.11±0.17 | 3.67±0.48 | 2.24±0.16 | 0.73±0.65 | 3.43±0.21 | 1.91±0.02 | 1.51±0.04 | 2.51 |
| *ExpoTab family* | | | | | | | | |
| ExpoTab($\alpha$=0.25) | 10.78±0.27 | 6.99±0.69 | 11.54±0.08 | 2.93±0.41 | 6.58±0.06 | 2.92±0.02 | 9.37±0.14 | 7.30 |
| ExpoTab($\alpha$=0.50) | 9.88±0.14 | 6.75±0.10 | 7.70±0.09 | 3.25±0.35 | 6.58±0.06 | 2.92±0.02 | 9.37±0.14 | 6.64 |
| ExpoTab($\alpha$=0.75) | 8.45±0.25 | 7.42±0.53 | 6.12±0.10 | 3.27±0.21 | 5.31±0.05 | 2.49±0.01 | 7.86±0.11 | 5.85 |
| ExpoTab($\alpha$=1.00) | 10.21±0.16 | 6.35±0.57 | 7.30±0.06 | 2.96±0.21 | 5.97±0.08 | 2.70±0.04 | 8.19±0.24 | 6.24 |

[1] TABSYN's performance is obtained via our reproduction.
[2] Results of other baselines (except Diabetes) are taken from Zhang et al. (2024).
[3] OOM entries are explained in the Computational Resources section.

Table 14: Comparison of $\alpha$-**Precision** scores. Bold Face

highlights the best score for each dataset. Higher scores reflect better performance.

| Method | Adult | Default | Shoppers | Magic | Beijing | News | Diabetes | Average |
|---|---|---|---|---|---|---|---|---|
| *GAN-based models* | | | | | | | | |
| CTGAN | 77.74±0.15 | 62.08±0.08 | 76.97±0.39 | 86.90±0.22 | 96.27±0.14 | 96.96±0.17 | 79.89±0.10 | 82.40 |
| *LLM-based model* | | | | | | | | |
| GReaT | 55.79±0.03 | 85.90±0.17 | 78.88±0.13 | 85.46±0.54 | 98.32±0.22 | OOM | OOM | 80.87 |
| *Diffusion-based models* | | | | | | | | |
| TabDDPM | 96.36±0.20 | 97.59±0.36 | 88.55±0.68 | 98.59±0.17 | 97.93±0.30 | 0.00±0.00 | 28.35±0.11 | 72.48 |
| TABSYN | 99.39±0.18 | 98.65±0.23 | 98.36±0.52 | 99.42±0.28 | 97.51±0.24 | 95.05±0.30 | 96.61±0.24 | 97.86 |
| CDTD | 99.38±0.25 | **99.36**±0.30 | 98.89±0.43 | 98.56±0.36 | **99.29**±0.36 | 95.75±0.29 | 98.72±0.18 | 98.56 |
| TabREP-DDPM | 99.11±0.25 | 98.52±0.36 | 96.33±0.34 | 98.52±0.36 | 98.98±0.16 | 95.35±0.11 | 99.08±0.13 | 97.98 |
| TabDiff | 99.02±0.20 | 98.49±0.28 | **99.11**±0.34 | **99.47**±0.21 | 98.06±0.24 | 97.36±0.17 | 95.69±0.19 | 98.17 |
| Tab-GeoDiff | **99.41**±0.27 | 99.21±0.16 | 99.15±0.25 | 99.01±0.08 | 99.05±0.19 | 98.83±0.28 | **99.18**±0.23 | **98.49** |
| *Flow-based models* | | | | | | | | |
| TabREP-FLOW | 98.21±0.34 | 96.50±0.22 | 95.85±0.46 | 98.79±0.25 | 98.16±0.13 | 90.91±0.25 | 99.08±0.13 | 96.49 |
| Tab-GeoFlow | 99.01±0.23 | 98.03±0.32 | 99.07±0.30 | 99.08±0.37 | 98.91±0.12 | **99.08**±0.12 | 99.15±0.30 | 98.11 |
| *ExpoTab family* | | | | | | | | |
| ExpoTab($\alpha$=0.25) | 96.50±0.24 | 94.52±0.20 | 79.28±0.48 | 94.65±0.43 | 97.40±0.28 | 89.82±0.23 | 77.72±0.21 | 89.99 |
| ExpoTab($\alpha$=0.50) | 91.63±0.34 | 91.04±0.31 | 92.22±0.64 | 94.94±0.34 | 95.65±0.29 | 97.01±0.43 | 88.89±0.41 | 92.91 |
| ExpoTab($\alpha$=0.75) | 99.19±0.32 | 90.82±0.23 | 96.78±0.36 | 98.65±0.43 | 98.47±0.24 | 76.02±0.33 | 98.16±0.31 | 92.96 |
| ExpoTab($\alpha$=1.00) | 85.96±0.24 | 98.58±0.28 | 99.12±0.25 | 92.89±0.43 | 96.10±0.31 | 87.92±0.43 | 80.56±0.32 | 91.33 |

[1] TabREP-FLOW and TabREP-DDPM's performance is obtained via our reproduction.
The results of other baselines except on Diabetes are taken from Zhang et al. (2024).
OOM entries are explained in the Computational Resources section.

Table 15: **DCR score**, representing the probability that a generated data sample is more similar to the training set than the test set. Bold Face highlights the best score for each dataset. A score closer to 50% is preferable.

| Method | Adult | Default | Shoppers | Magic | Beijing | News | Diabetes | Average |
|---|---|---|---|---|---|---|---|---|
| *Diffusion-based models* | | | | | | | | |
| CDTD | 0.7562±0.0021 | **0.8990**±0.0023 | 0.8998±0.0025 | **0.8975**±0.0037 | **0.8995**±0.0034 | 0.9002±0.0012 | **0.7980**±0.0011 | 0.8643 |
| TabREP-FLOW | 0.8980±0.0019 | 0.9060±0.0016 | 0.8980±0.0019 | 0.9205±0.0027 | 0.8998±0.0019 | 0.9010±0.0032 | 0.8038±0.0034 | **0.8179** |
| TabREP-DDPM | 0.6846±0.0027 | 0.9034±0.0016 | 0.9160±0.0032 | 0.9034±0.0016 | 0.9034±0.0016 | **0.8073**±0.0030 | 0.8900±0.0230 | 0.8583 |
| *Geo-based models* | | | | | | | | |
| Tab-GeoDiff | 0.6738±0.0025 | 0.9009±0.0017 | 0.9044±0.0026 | 0.9013±0.0082 | 0.9105±0.0010 | 0.9034±0.0015 | 0.8019±0.0014 | 0.8566 |
| Tab-GeoFlow | 0.6753±0.0024 | 0.9037±0.0018 | 0.9101±0.0025 | 0.9585±0.0037 | 0.9100±0.0011 | 0.9005±0.0018 | 0.8018±0.0010 | 0.8657 |
| ExpoTab($\alpha$=0.25) | 0.6655±0.0029 | 0.9035±0.0021 | 0.9026±0.0016 | 0.9005±0.0018 | 0.8998±0.0012 | 0.8996±0.0012 | 0.8042±0.0012 | 0.8380 |
| ExpoTab($\alpha$=0.50) | 0.6651±0.0031 | 0.9026±0.0031 | 0.9016±0.0042 | 0.9021±0.0033 | 0.9006±0.0022 | 0.8997±0.0012 | 0.8028±0.0032 | 0.8392 |
| ExpoTab($\alpha$=0.75) | 0.6669±0.0024 | 0.9024±0.0031 | 0.9048±0.0028 | 0.9018±0.0014 | 0.9019±0.0034 | 0.8989±0.0032 | 0.8027±0.0022 | 0.8400 |
| ExpoTab($\alpha$=1.00) | **0.6618**±0.0029 | 0.9035±0.0021 | **0.8921**±0.0031 | 0.9019±0.0022 | 0.9002±0.0042 | 0.8990±0.0032 | 0.8031±0.0015 | 0.8359 |

[1] TABSYN's performance is obtained via our reproduction. The results of other baselines except on Diabetes are taken from [

Table 16: Comparison of generative models under fixed-compute and extended-training regimes. Best results are highlighted in **blue**. NA indicates cases where models failed to produce meaningful results.

| Sampling Method | Model | $\alpha$-Precision (↑) | | | | | | Machine Learning Efficiency (MLE) | | | | | |
|---|---|---|---|---|---|---|---|---|---|---|---|---|---|
| | | Adult | Beijing | News | Default | Diabetes | Shoppers | Adult | Beijing | News | Default | Diabetes | Shoppers |
| One-step sampling | CDTD | 0.13±0.32 | 0.00±0.00 | 6.67±0.15 | 6.66±0.11 | NA | 6.66±0.12 | NA | 1.014±0.224 | 0.892±0.003 | NA | NA | NA |
| | TabREP-Flow | 56.08±0.21 | 38.38±0.18 | 6.75±0.17 | 18.14±0.09 | 6.89±0.31 | 16.59±0.22 | NA | 1.043±0.271 | 0.880±0.125 | NA | 0.472±0.004 | NA |
| | TabREP-DDPM | 2.04±0.29 | NA | NA | 0.17±0.01 | NA | 0.77±0.09 | 0.343±0.003 | 2.293±0.195 | 3.702±0.174 | 0.487±0.002 | 0.485±0.002 | 0.503±0.002 |
| | Tab-GeoFlow | 40.18±0.13 | 41.36±0.27 | 12.49±0.14 | 43.86±0.31 | 9.64±0.05 | 20.66±0.26 | 0.510±0.001 | 1.013±0.054 | 0.863±0.011 | NA | NA | NA |
| | Tab-GeoDiff | 2.55±0.17 | 0.01±0.00 | NA | 0.04±0.00 | NA | 0.82±0.01 | 0.490±0.003 | 3.101±0.110 | 3.244±0.412 | 0.505±0.002 | 0.507±0.001 | 0.458±0.002 |
| MeanFlow-based | Tab-MeanFlow | 97.36±0.11 | **98.88±0.12** | **98.51±0.19** | 86.98±0.28 | 77.14±0.21 | 93.82±0.66 | 0.893±0.007 | **0.695±0.024** | **0.843±0.013** | 0.739±0.011 | 0.590±0.016 | 0.895±0.005 |
| **Ours** | ExpoTab | **99.23±0.31** | 98.47±0.24 | 97.01±0.43 | **98.58±0.28** | **98.16±0.31** | **99.12±0.25** | **0.900±0.011** | 0.701±0.014 | 0.866±0.020 | **0.748±0.012** | **0.628±0.007** | **0.902±0.005** |
| **Improvement** | | 1.92% ↑ | 0.41% ↑ | 1.52% ↓ | 13.34% ↑ | 27.25% ↑ | 5.65% ↑ | 0.78% ↑ | 0.86% ↓ | 2.73% ↓ | 1.22% ↑ | 6.44% ↑ | 0.78% ↑ |

Table 17: Comparison of generative models. Best results are in **blue**. OOM means Out of Memory.

| METHODS | α-PRECISION (↑) | | | | | | SAMPLING TIME(S) (↓) | | | | | |
|---|---|---|---|---|---|---|---|---|---|---|---|---|
| | ADULT | BEIJING | NEWS | DEFAULT | DIABETES | SHOPPERS | ADULT | BEIJING | NEWS | DEFAULT | DIABETES | SHOPPERS |
| CTGAN | $77.74_{\pm0.15}$ | $96.27_{\pm0.14}$ | $96.96_{\pm0.17}$ | $62.08_{\pm0.08}$ | $79.89_{\pm0.10}$ | $76.97_{\pm0.39}$ | 0.86 | 0.93 | 2.36 | 1.08 | 10.08 | 0.41 |
| TabDDPM | $96.36_{\pm0.20}$ | $97.93_{\pm0.30}$ | $0.00_{\pm0.00}$ | $97.59_{\pm0.36}$ | $28.35_{\pm0.11}$ | $88.55_{\pm0.68}$ | 44.96 | 40.01 | 33.26 | 32.46 | 84.36 | 23.22 |
| TabSyn | $99.39_{\pm0.18}$ | $97.51_{\pm0.24}$ | $95.05_{\pm0.30}$ | $98.65_{\pm0.23}$ | $96.61_{\pm0.24}$ | $98.36_{\pm0.52}$ | 2.43 | 2.56 | 5.25 | 2.42 | 7.89 | 1.00 |
| CDTD | $99.37_{\pm0.27}$ | $99.29_{\pm0.36}$ | $95.75_{\pm0.29}$ | $99.36_{\pm0.31}$ | $98.72_{\pm0.18}$ | $98.89_{\pm0.43}$ | 5.43 | 3.53 | 5.35 | 5.19 | 21.89 | 2.02 |
| TabDiff | $99.02_{\pm0.20}$ | $98.06_{\pm0.24}$ | $97.36_{\pm0.17}$ | $98.49_{\pm0.28}$ | $95.69_{\pm0.19}$ | $99.11_{\pm0.34}$ | 10.27 | 6.86 | 9.21 | 7.26 | 29.23 | 4.65 |
| TabREP-Flow | $98.21_{\pm0.34}$ | $98.16_{\pm0.13}$ | $90.91_{\pm0.25}$ | $96.50_{\pm0.44}$ | $99.08_{\pm0.13}$ | $95.85_{\pm0.46}$ | 1.52 | 1.33 | 1.39 | 1.01 | 2.77 | 0.50 |
| TabREP-DDPM | $99.11_{\pm0.15}$ | $98.98_{\pm0.16}$ | $95.35_{\pm0.11}$ | $98.66_{\pm0.24}$ | $97.19_{\pm0.22}$ | $96.14_{\pm0.19}$ | 35.11 | 26.44 | 28.18 | 21.14 | 54.84 | 10.33 |
| **Tab-GeoFlow(Ours)** | $99.01_{\pm0.23}$ | $98.91_{\pm0.12}$ | $\mathbf{99.08_{\pm0.13}}$ | $98.03_{\pm0.32}$ | $\mathbf{99.15_{\pm0.24}}$ | $99.07_{\pm0.30}$ | 2.89 | 2.61 | 2.63 | 1.97 | 5.44 | 0.93 |
| **Tab-GeoDiff(Ours)** | $\mathbf{99.41_{\pm0.37}}$ | $99.05_{\pm0.19}$ | $98.83_{\pm0.27}$ | $99.21_{\pm0.16}$ | $97.79_{\pm0.35}$ | $\mathbf{99.15_{\pm0.25}}$ | 26.64 | 26.12 | 27.26 | 21.31 | 54.99 | 10.64 |
| **ExpoTab(Ours)** | $99.23_{\pm0.31}$ | $98.47_{\pm0.24}$ | $97.01_{\pm0.43}$ | $98.58_{\pm0.28}$ | $98.16_{\pm0.31}$ | $99.12_{\pm0.25}$ | **0.20** | **0.22** | **0.31** | **0.21** | **0.36** | **0.19** |
| IMPROV. (EXPOTAB) | 0.18%↓ | 0.58%↓ | 2.01%↓ | 0.78%↓ | 0.99%↓ | 0.03%↓ | 76.74%↓ | 76.34%↓ | 77.70%↓ | 79.21%↓ | 87.00%↓ | 53.66%↓ |

Table 18: Comparison of Machine Learning Performance measured in AUC (%) (higher is better) and Distance to Closest Record (DCR)(lower is better). Best results are in **black**.

| METHODS | MACHINE LEARNING PERFORMANCE (↑) | | | | DISTANCE TO CLOSEST RECORD (↓) | | | |
|---|---|---|---|---|---|---|---|---|
| | BIORESPONSE | GINA | ISOLET | HAR | BIORESPONSE | GINA | ISOLET | HAR |
| CTGAN | $41.45_{\pm0.23}$ | $42.93_{\pm0.36}$ | $22.62_{\pm0.11}$ | $68.19_{\pm0.36}$ | $\mathbf{0.892_{\pm0.003}}$ | $0.895_{\pm0.007}$ | $\mathbf{0.809_{\pm0.009}}$ | $\mathbf{0.893_{\pm0.004}}$ |
| TabDDPM | $70.02_{\pm0.45}$ | $56.94_{\pm0.21}$ | $85.74_{\pm0.24}$ | $74.67_{\pm0.52}$ | $0.919_{\pm0.016}$ | $0.901_{\pm0.011}$ | $0.926_{\pm0.009}$ | $0.937_{\pm0.006}$ |
| CDTD | $81.82_{\pm0.41}$ | $\mathbf{91.51_{\pm0.29}}$ | $98.29_{\pm0.14}$ | $99.72_{\pm0.39}$ | $0.906_{\pm0.009}$ | $0.913_{\pm0.011}$ | $0.911_{\pm0.005}$ | $0.907_{\pm0.005}$ |
| TabREP-Flow | $65.86_{\pm0.18}$ | $80.67_{\pm0.31}$ | $97.92_{\pm0.27}$ | $96.56_{\pm0.21}$ | $0.923_{\pm0.005}$ | $0.917_{\pm0.004}$ | $0.908_{\pm0.006}$ | $0.954_{\pm0.005}$ |
| TabREP-DDPM | $79.26_{\pm0.22}$ | $90.34_{\pm0.26}$ | $98.23_{\pm0.21}$ | $96.79_{\pm0.24}$ | $0.927_{\pm0.004}$ | $0.908_{\pm0.003}$ | $0.916_{\pm0.005}$ | $0.941_{\pm0.004}$ |
| **Tab-GeoFlow(Ours)** | $69.08_{\pm0.20}$ | $83.83_{\pm0.23}$ | $98.64_{\pm0.42}$ | $\mathbf{99.84_{\pm0.17}}$ | $0.909_{\pm0.003}$ | $0.915_{\pm0.004}$ | $0.922_{\pm0.004}$ | $0.927_{\pm0.003}$ |
| **Tab-GeoDiff(Ours)** | $72.56_{\pm0.18}$ | $83.42_{\pm0.21}$ | $98.15_{\pm0.33}$ | $97.78_{\pm0.16}$ | $0.902_{\pm0.002}$ | $\mathbf{0.894_{\pm0.002}}$ | $0.895_{\pm0.003}$ | $0.909_{\pm0.002}$ |
| **ExpoTab(Ours)** | $\mathbf{82.06_{\pm0.19}}$ | $90.39_{\pm0.22}$ | $\mathbf{99.96_{\pm0.21}}$ | $98.96_{\pm0.15}$ | $0.916_{\pm0.012}$ | $0.903_{\pm0.004}$ | $0.912_{\pm0.007}$ | $0.918_{\pm0.010}$ |
| IMPROV. (EXPOTAB) | 2.64%↑ | 2.58%↑ | 3.39%↑ | 2.91%↑ | 15.64%↓ | 16.09%↓ | 15.65%↓ | 15.66%↓ |

