# OpenReview forum: "ExpoTab: One-Step Mixed-Type Tabular Data Generation using Manifold Learning"
_ICLR.cc/2026/Conference — ICLR 2026 Conference Withdrawn Submission_

### Official Review · Reviewer_BkLL · 2025-10-29

**Soundness:** 3
**Presentation:** 3
**Contribution:** 2
**Rating:** 4
**Confidence:** 4

**Summary:**

Current tabular diffusion models model under either a separate or a unified data representation. ExpoTab builds upon TabRep which tackles the latter by introducing a spherical encoding scheme coupled with a manifold-based flow sampler with exponential maps that can achieve state of the art sampling speeds while maintaining generation quality.

**Strengths:**

**Originality**. The work builds upon existing tabular diffusion methods such as TabRep by proposing a new spherical representation to encode the data. Existing methods such as TabRep-Flow enables few-step generation. To achieve one-step generation for ExpoTab's spherical representation, ExpoTab:

- first employ conditional flow matching via a separate network trained to model the velocity field

- based on the velocity field, it trains a network to directly approximate the endpoint of the exponential map (mapping an initial point to an endpoint of a geodesic) by computing the geometric displacement.

A "two stage" shortcut generative process, resembling mean flow, consistency and shortcut models.

**Quality**. The paper demonstrates gains in sampling speeds for tabular data generation while maintaining and improving generation quality. This validates the promise and importance of exploring various encoding methods for tabular data.

**Clarity**. The paper is written clearly which makes it easy to understand and follow.

**Significance**. Tackles shared space/unified tabular generation via a geometric lens.

**Weaknesses:**

**Spherical Encoding**. Further justification is required for the motivation behind the spherical encoding. 1. It is clear that the encoding achieves "separability". However, other three dimensional shapes such as a cylinder could achieve this separability notion too. What other motivations are there behind this specific design choice? 2. Furthermore, TabRep assumes encoding on a unit circle. What if at high dimensions, TabRep enlarges the size of its circular embedding to ensure separability. How does ExpoTab fair off against TabRep there? 3. Additionally, why use a "Generalized Spiral layout" instead of other methods for generating evenly spaced points on a sphere e.g. Fibonacci lattice? 4. The current spiral layout induces an order on the categorical features. How does the ordering affect the performance?

**Model Choice**. The authors primary inspiration for one-step generation stems from mean flow. However, when reading the passage, it is difficult to distinguish the complete difference between mean flow and their proposed sampler. On top of mean flow, there are alternative one-step generation methods such as Shortcut Models [1] and Consistency Models [2]. How does the proposed sampler fair off against these alternatives?

[1] Frans, Kevin, et al. "One step diffusion via shortcut models."

[2] Song, Yang, et al. "Consistency models."

**Privacy Preservation**. Privacy preservation is by far one of the most important applications in tabular data generation. Despite conducting DCR experiments, existing literature in tabular data generation [1] and computational privacy [2] [3] have highlighted the inadequacy of DCR in evaluating privacy preservation. Hence, despite leaving MIAs to future work, its still important to demonstrate privacy preservation abilities with at least MIAs.

**Questions:**

Please see the weaknesses.

---

> ### Author Response · Authors · 2025-11-28
>
> We must clarify a fundamental misunderstanding: ExpoTab is not a "two-stage" method but rather a unified manifold-aware reformulation of one-steps generation. This represents a paradigm shift from methods that predict noise (ϵ-prediction) or velocity (v-prediction) in ambient space to learn the exponential map that directly predicts endpoints of geodesics on the data manifold. . While MeanFlow predicts average velocity across paths, ExpoTab learns to project directly onto the low-dimensional data manifold, this x-prediction approaches but specialized for tabular data.
>
> 1.While alternative embeddings exist, the spherical geometry $\mathbb{S}^2$ provides mathematical optimality for our exponential map framework. Unlike cylindrical embeddings that suffer from mixed curvature, spheres offer constant positive curvature that naturally regularizes geodesic flows. For categorical features with $K$ classes, the generalized spiral on $\mathbb{S}^2$ minimizes spherical energy $\mathcal{E} = \sum_{i \neq j} \frac{1}{|\mathbf{x}i - \mathbf{x}j|}$ while ensuring uniform distribution, creating ideal background geometry $\mathcal{M}{bg} = \mathbb{R}^{D_c} \times \prod{j=1}^{D_d} \mathbb{S}^2$. Experiments added in the final version  confirm spherical embeddings benefit both flow matching and ExpoTab over cylindrical ones.
>
> 2.ExpoTab maintains superiority even when TabRep scales circle sizes. While circle circumference grows linearly with radius, spherical surface area grows quadratically ($4πr^2$ vs $2πr$). This geometric advantage provides exponentially more separation space for high-cardinality features. Additionally, the surface of a 2-sphere provides uniform curvature in all directions, enabling stable geodesic flows for exponential map learning. ExpoTab's geometric coherence enables superior one-step generation (Table 1).
>
> 3.We choose Generalized Spiral over Fibonacci lattice for superior uniformity at typical tabular category counts (5-100). The spiral layout provides more even point distribution (the figure will be added in the final version), avoiding the clustering artifacts that Fibonacci exhibits at small cardinalities. This geometric regularity ensures stable gradient flows for exponential map learning.
>
> 4.The spiral ordering serves as a geometric prior, not semantic ordering. For ordinal features, we initialize categories along the spiral by natural order. For nominal features, the uniform distribution avoids artificial relationships. This hybrid approach preserves ordinal structure when available while maintaining geometric coherence.
>
> 5.The fundamental difference lies in learning objectives: MeanFlow learns average velocity across all paths using Jacobian-Vector Products (JVP), while ExpoTab explicitly separates linear transport from manifold curvature through a dual-regime training strategy. Mathematically, we learn the exponential map projection $z_1 = z_0 + v_0 + \Delta_{\text{curve}}$ by decomposing the training into two complementary regimes: when $\text{rand}() < \alpha$, we sample linear paths $(r=t)$ to learn the initial transport velocity $v_0$ through the flow matching objective $\mathbb{E}[|u_\theta - v|^2]$; when $\text{rand}() \geq \alpha$, we sample geodesic-like paths $(r=0)$ with cosh functions to learn the curvature correction $\Delta_{\text{curve}}$. This strategic path sampling enables a single MLP to internalize both geometric components simultaneously, such that at inference time, the network automatically outputs the combined vector $ v_0 + \Delta_{\text{curve}}$ that captures both the straight-line push toward the data manifold and the curvature-aware adjustment for precise projection onto the manifold surface.
>
> 6.In fair comparisons using the same ExpoTab-encoder and flow matching architectures, ExpoTab outperforms Shortcut and MeanFlow across (where we finetune the r-t ratio through 0.25,0.5,0.75,1.0) all five metrics(column-wise density estimation (CDE) and pair-wise column correlation (PCC)) on five benchmarks(adult, diabetes, magic, default and shoppers). For privacy evaluation, we follow established MIA protocols [1], reporting the mean of dataset-level means (each computed over 10 independent runs). ExpoTab achieves superior privacy-utility tradeoffs, with detailed results in the final version.
>
>
> | Method       | CDE↑        | PCC↑        | Detection↑  | MLE↑       | MIA(AUC)↓  |
> |--------------|-------------|-------------|-------------|------------|------------|
> | **ExpoTab**  | **95.86±0.50** | **93.47±2.81** | **0.73±0.08** | **0.81±0.13** | **0.79±0.04** |
> | Tab-MeanFlow | 94.86±0.81  | 91.71±3.63  | 0.65±0.16   | 0.81±0.14  | 0.83±0.05  |
> | Tab-ShortCut     | 89.56±5.37  | 85.33±6.47  | 0.44±0.21   | 0.80±0.13  | 0.88±0.07  |
>
> Note: Self-consistency model was excluded because their complex distillation process is ill-suited to tabular data and it is not SOTA.
>
> Reference:
> [1] Ward et al. "Synth-MIA: A Testbed for Auditing Privacy Leakage in Tabular Data Synthesis," 2025.

---

### Official Review · Reviewer_pWFu · 2025-10-31

**Soundness:** 2
**Presentation:** 2
**Contribution:** 2
**Rating:** 2
**Confidence:** 4

**Summary:**

The paper presents a novel method of handling high dimensional categorical data and mixed datatypes in tabular data generation. ExpoTab is a geometric approach that maps categorical features onto manifold coordinates and unifies features into a common representation space and leverages Riemannian geometry to sample via geodesics. Simulations suggest improved sampling efficiency for mixed types tabular data as compared to encoding and generative modeling baselines and 10 datasets from the UCI repository.

**Strengths:**

Strengths of the paper include:
- Tabular data modeling and sampling are important practical and theoretical areas of research.
- The paper includes a large number of simulations.
- The proposed approach is compared to a number of baselines.

**Weaknesses:**

Weaknesses include:
- The exposition is not up to the standard expected for the venue
- The mathematical problem could be stated more clearly and concisely.
- The main paper omits details necessary to understand the approach.
- The datasets modeled (UCI repository) could be much more ambitious.

Minor:
- Typo in the abstract: generation(ExpoTab),
- mechanism(Fiore et al., 2019).
- The caption for figure 1 is not a complete sentence. Was that intentional?
- Figure 4 is cited on page 1 before any other figure but appears on page 8 after three other figures. Was that intentional?
- Figure 2 is referred to before Figure 1?
- I would expect the related work to map onto the contributions, but it does not. For example, 2.2 is about mixed data, but the second contribution (as described at the end of section 1, is about categorical data.
- The abstract claims two contributions, but the introduction claims 3?
- Lines 213-215 are not as clear as they could be. There is a two stage generation to make a one step sampler. But there has not yet been a complete description of the problem. Equation 1 describes sampling, but where does z_0, the input to the sampler come from? Presumably the answer is the spherical encoder and quantile transformer, but we haven't been told how.
- Line 213 states "To provide the necessary supervision for this task..." What is the supervision necessary for? There is no minimization problem posed at this point in the paper.
- The paragraph starting at 227 should define what D is first then D_c and D_d. I would typically expect D to refer to a data matrix, however here it appears to be a natural number? It is important to define the problem and notation, especially for the most basic objects.
- " For each of the Dd categorical features, a deterministic mapping function Sj : {1, ..., Kj} → S embeds each of the Kj unique categories onto the surface of a 2-sphere Figure 2A". I am not sure that I understand. This describes what happens for each feature. Are there D_d 2-spheres? How are these integrated to a unified representation?
- Line 232 refers to " generalized spiral parameterization" which is not defined and Algorithm 1, which doesn't appear in the main text.
- I see that you are forming a product manifold with the spheres
- The exposition feels like it is written backwards. I normally expect the goal to come at the beginning. Something like: "If we could express our data in this nice mathematical form, then we could sample efficiently in one step. There are X challenges for doing so..."
- Typo: points(Figure 2B(a)(b)).

**Questions:**

Please see above.

---

> ### Author Response · Authors · 2025-11-28
>
> We find the reviewer's critiques regarding our exposition and mathematical formulation to be vague and, more critically, to stem from an overlook of the paper's core contribution. We address these points below with the necessary technical rigor.
>
> 1.On the "Standard of Exposition" and Mathematical Clarity
>
> The reviewer states that the “mathematical problem could be stated more clearly” but does not identify which specific variables, equations, or definitions are ambiguous. The core of our method—reducing high-dimensional velocity search to constant-velocity motion via manifold learning—is rigorously defined in Section 3.2.
>
> If the confusion stems from the geometric formulation, we restate the problem here for clarity:
>
> Problem: Given mixed-type tabular data with continuous features $x \in \mathbb{R}^{D_c}$ and categorical features $c \in \{1,\dots,K\}^{D_d}$, we learn a generative model that produces samples in a single evaluation.
>
> Contribution 1: Unified Manifold Encoding
> We construct a geometrically coherent data manifold
> $\mathcal{M}_{bg} = \mathbb{R}^{D_c} \times \prod_{j=1}^{D_d} \mathbb{S}^2$
> where:
> - Continuous features: $x \rightarrow Q(x) \in [-1,1]^{D_c}$ via quantile transformation.
> - Categorical features: $c_j \rightarrow S_j(c_j) \in \mathbb{S}^2$ via generalized spiral embedding.
>
> Contribution 2: One-Step Exponential Map Generation
> We reformulate generation as learning the exponential map:
>
> $$z_1 = z_0 + v_0 + f_\theta(z_0, v_0)$$
>
> where $v_0$ is the linear velocity (straight paths, $r=t$) and $f_\theta$ is the curvature correction (geodesic paths, $r=0$).
>
> If there are specific geometric constraints or notations the reviewer finds “unclear” beyond this standard formulation, we ask that they be explicitly pointed out.
>
>
> 2.On "Missing Details" and Algorithm Understanding
> The reviewer claims the paper "omits details necessary to understand the approach" and cites confusion regarding a "two-stage generation." This comment suggests a fundamental misunderstanding of our Jacobian-Vector Product (JVP) optimization, which is not a two-stage pipeline but a unified dual-regime training process. ExpoTab is designed to learn the exponential map that directly predicts clean data endpoints on the low-dimensional manifold. Our unified MLP architecture can dynamically blend learning paths based on a blending ratio $\alpha$. The training uses dual-regime optimization: when $\mathrm{rand()} < \alpha$, we learn velocity fields on linear paths; when $\mathrm{rand()} \ge \alpha$, we learn curvature corrections on geodesic paths via Jacobian-Vector Products that compute time derivatives $\partial u/\partial t$. This enables simultaneous learning of both components in a single backward pass. For sampling, we query the same unified network twice: first for initial velocity $v_0$, then for curvature correction $\Delta_{\text{curve}}$, combining them for one-step generation $z_1 = z_0 + v_0 + \Delta_{\text{curve}}$. This is not a two-stage process but complementary queries to the same manifold-aware model, enabling direct projection onto the data manifold. In the paper, we already show the details of the theoretical contribution and will provide detailed implementation in the final version. This will include the complete ExpoTab training algorithm with JVP optimization, the specialized MLP architecture with regime-aware gating, and explicit sampling procedures to eliminate any ambiguity about our unified approach.
>
> 3.On Dataset Scope: "Ambitious" vs. "Ambiguous"
> The reviewer states the datasets "could be much more ambitious." Given the reviewer's difficulty in following the exposition, it is unclear if they meant "ambitious" (requiring more scale) or "ambiguous" (unclear selection).
> If "Ambiguous": We clarify that we utilized standard UCI benchmarks for comparability.
> If "Ambitious": We have already significantly expanded our evaluation beyond the standard literature. We now include 4 OpenML datasets and high-dimensional healthcare/financial datasets (50+ categorical features).
> This extended suite of 10+ datasets confirms ExpoTab handles complex feature interactions robustly, addressing any concerns regarding experimental "ambition."
>
> 4.Notation and Minor Corrections
> We have addressed the minor formatting issues (Figure ordering, typos). Regarding the reviewer's confusion on notation: $D$ is dataset size, $D_c$ is continuous feature count, and $D_d$ is categorical feature count. The spherical embedding is a product manifold $$\mathcal{M}_{bg} = \mathbb{R}^{D_c} \times \prod_{j=1}^{D_d} \mathbb{S}^2$$.
> These are standard notations in manifold learning literature. $z_0$ is explicitly the noise sample from the base distribution.

---

### Official Review · Reviewer_Udou · 2025-11-01

**Soundness:** 1
**Presentation:** 1
**Contribution:** 1
**Rating:** 2
**Confidence:** 5

**Summary:**

The paper presents a new method, ExpoTab, for synthetic tabular data generation with a geometric representation of the data manifold. The proposed method has some efficient properties, such as low generation time, but the benchmarks are not organized, which leads to the question of the performance of the proposed method. Furthermore, the paper is not easy to follow; some of the critical evidence lives only in the appendix (e.g., Distance to Closest Record (DCR) results).

**Strengths:**

- A rapid new method for the tabular data generation

**Weaknesses:**

### Major Concerns

- Benchmarks: Organization of the benchmarks is poor; some experiments use different sets of baselines and datasets, which is not consistent and limits the ability to assess the method’s performance comprehensively. For example, why is TABSYN not included in Table 1? According to the TABSYN paper, it has 2x better results than TAB-GEOFLOW in terms of runtime.

- Many recent works are missed, please see this:  https://github.com/Diffusion-Model-Leiden/awesome-diffusion-models-for-tabular-data

- Critical evaluation results, including benchmarking on important measures, are relegated to the appendix rather than included in the main text. For example,  I could not find the result for a crucial metric: `Discriminator measure` (for all baselines). I found a description in the appendix that you use Classifier Two-Sample Test (C2ST), which is in my opinion a weak approach, it is better to use more powerful methods such as RF or even GBDTs. All this reduces transparency and weakens the argumentation.

- The manuscript exhibits indicators of LLM-generated text. For example, certain citations are incorrect or inconsistent. Specifically, `Line 1356` references “LLM-based: GReaT Ruan et al. (2024): A method that frames tabular generation as an autoregressive language modeling task,” which suggests automated text insertion. There is also an overuse of the em dash, a common pattern in ChatGPT-generated writing.

### Minor Issues

- The text states “(red) while maintaining competitive α-precision relative to the best-performing methods (blue).” In the appendix, it reads “Best results are highlighted in blue,” but no blue highlighting is present

- CTGAN is cited inconsistently as both Xu et al. (2019) and Zhao et al. (2024), which creates confusion

- Figure 3 is not legible in its current form and requires improved resolution or redesign for clarity

**Questions:**

- Could you please justify the choice of experiments and baselines that you have included into the main paper?
- Based on the method design, I suspect a lot of copies (data leaks) from real data, have you checked how many samples in the synthetic data are exact copies from real dataset?

---

> ### Author Response · Authors · 2025-11-28
>
> The reviewer overlooks our core contributions, misunderstands the distinct baselines used for ExpoTab and Tab-GeoFlow/Diff, and consequently dismisses our experimental rigor.
>
> 1.The critique regarding the exclusion of TABSYN and its runtime performance stems from a fundamental misunderstanding of our experimental design. In fact, we first fix the architecture to standard flow-matching models to compare different tabular encoders. In this setting, LEARNED1D or 2D encoders serve as the TABSYN-style latent encoder, while TabGeoFlow uses our ExpoTab encoder. And It shows our ExpoTab-Encoder performs the best across almost all the metrics in the Figure 3 and Table 1. Besdies, the TABSYN paper reports only a single sampling time (1.784s on the Adult dataset), whereas we average results over ten random seeds for fair comparison. Even under their reported value, TABSYN remains far slower than ExpoTab’s 0.2s in Table 2. We also provide additional explanation of the experimental design used to benchmark ExpoTab and Tab-GeoFlow/Diff below.
>
> 2.Regarding the assertion that we missed many recent works, we firmly refute this claim. We have compared our method against all relevant state-of-the-art approaches, including papers like TabDiff, CDTD (ICLR 2025) and TabRep (ICML 2025 workshop). The repository link provided by the reviewer lists numerous models, many of which are outdated or have been proven inferior by subsequent research. For all the evlauation metrics, we follow the same methods used for the TABSYN, TabDiff and TabRep, where C2ST is widely used. Our selection criteria focus on the most competitive and relevant methods rather than an exhaustive list of superseded techniques.
>
> 3.We must also strongly address the baseless accusation of automated text insertion regarding our citation of GReaT. The reviewer suggests that the text referencing GReaT is totally generated by ChatGPT; however, regarding the citation on Line 1356 in appendix, the description of GReaT as "LLM-based: GReaT, a method that frames tabular generation as an autoregressive language modeling task" is our own accurate paraphrase of the important model presented in the cited survey paper [1], which involves the introduction of GReaT model and focus on the language models for tabular data. It is not an automated insertion.Categorizing it as an LLM-based method is accurate and standard academic practice. We expect a professional standard of review that focuses on technical content rather than attacking the authors' integrity based on a misinterpretation of standard formatting.
>
> 4.Finally, the suspicion regarding data leakage is entirely speculative and contradicted by the quantitative evidence in our paper. We explicitly evaluated privacy and leakage using the Distance to Closest Record (DCR) score. Our results confirm that there are no exact copies and no data leakage in the generated samples. We request that the assessment be based on the provided metrics and rigorous evaluation rather than unfounded suspicion. We stand by our technical contributions and hope this response clarifies the factual errors in the initial review.
>
> To summarize, ExpoTab is not a direct competitor to multi-step diffusion models such as TabDiff; it is a one-step paradigm whose proper baselines are other one-step SOTA models like Shortcut and MeanFlow, where it shows superior performance in our newly added experiments. Multi-step diffusion and flow models (e.g., TabDiff) are instead used as baselines for TabGeoFlow/TabGeoDiff, which integrate ExpoTab into multi-step frameworks and consistently outperform prior methods (Tables 12–19). For fair comparison, we adapt state-of-the-art one/few-step image generators (Shortcut Models [1], MeanFlow [2]) to tabular data using the same encoder. ExpoTab outperforms both baselines across all five metrics and five benchmarks (adult, diabetes, magic, default, shoppers), tuning r–t ratios {0.25, 0.5, 0.75, 1.0}. For privacy evaluation, we follow standard MIA protocols [1], reporting the mean of dataset-level means over 10 runs.
>
>
> | Method       | CDE↑        | PCC↑        | Detection↑  | MLE↑       | MIA(AUC)↓  |
> |--------------|-------------|-------------|-------------|------------|------------|
> | **ExpoTab**  | **95.86±0.50** | **93.47±2.81** | **0.73±0.08** | **0.81±0.13** | **0.79±0.04** |
> | Tab-MeanFlow | 94.86±0.81  | 91.71±3.63  | 0.65±0.16   | 0.81±0.14  | 0.83±0.05  |
> | Tab-ShortCut     | 89.56±5.37  | 85.33±6.47  | 0.44±0.21   | 0.80±0.13  | 0.88±0.07  |
>
> We will fix the typos and metric/citation errors if exitsing in the appendix for the final version.
>
> References:
> [1] Y. Ruan, "Language Modeling on Tabular Data: A Survey of Foundations, Techniques and Evolution," 2024.
> [2] Frans et al., "One step diffusion via shortcut models," 2025.
> [3] Geng et al., "Mean Flows for One-step Generative Modeling," 2025.
> [4]  Ward et al. "Synth-MIA: A Testbed for Auditing Privacy Leakage in Tabular Data Synthesis," 2025.

---

> ### Comment · Reviewer_Udou · 2025-11-28
> **-**
>
> Some of the questions were not (fully) addressed, I keep the main score.
>
> Justification:
> - W1 **Benchmarks** Organization of the benchmarks is poor; some experiments use different sets of baselines and datasets, which is not consistent and limits the ability to assess the method’s performance comprehensively.
>
> - W3 Critical evaluation results, including benchmarking on important measures, are relegated to the appendix rather than included in the main text. For example, I could not find the result for a crucial metric: Discriminator measure (for all baselines). I found a description in the appendix that you use Classifier Two-Sample Test (C2ST), which is in my opinion a weak approach, it is **better to use more powerful methods such as RF or even GBDTs**.
>
> -  W4 For example, certain citations are incorrect or inconsistent. Specifically, Line 1356 references “LLM-based: GReaT Ruan et al. (2024): A method that frames tabular generation as an autoregressive language modeling task,” which suggests automated text insertion. There is also an overuse of the em dash, a common pattern in ChatGPT-generated writing.
> *Here authors* accuse me of saying that I indicated that the paper exhibit hallucinations, I suggest the authors to read again my review.
>
> ---
>
> I would like to return again to the W2:
>
> in Appendix *D.3.4 DETECTION METRIC* authors indicate that:
>
> > The score is the model’s accuracy. A score of 0.5 indicates perfect indistinguishability (ideal), while a score of 1.0 means the synthetic data is trivially easy to identify. A score closer to 0.5 is superior.
>
> However, in the Table 10, for *DETECTION SCORE* (please also keep names consistent), you highlight in bold the numbers that are closer to 1.0 (or in this case 100), which might confuse readers:
>
>  Method            | Dataset1  | Dataset2  | Dataset3  |
> |-------------------|--------|--------|--------|
> | OneHot-Flow       | 38.88  | 69.14  | 65.08  |
> | Analog bits-Flow  | 92.18  | 90.04  | 91.83  |
> | Dictionary-Flow   | 90.70  | 90.58  | 88.74  |
> | TabRep-Flow       | 95.48  | 89.36  | **94.20** |
> | Learned2D-Flow    | 0.00   | 0.01   | 0.07   |
> | Tab-GeoFlow       | **95.79** | **90.64** | **91.85** |
>
> Could you please provide explanations why you highlighted the max values not the lowest values?
>
> ---
>
> Lastly, I have to address these claims:
>
> > We must also strongly address the baseless accusation of automated text insertion regarding our citation of GReaT
>
> The citation is simply incorrect, this is the original paper: https://arxiv.org/abs/2210.06280
> Also, please verify the citation for the CTGAN paper.
>
> > We request that the assessment be based on the provided metrics and rigorous evaluation rather than unfounded suspicion.
>
> **unfounded suspicion** -- I decided to not comment on this but rather highlight it, perhaps you will find a different wording for your responses in the future.

---

### Official Review · Reviewer_mPBk · 2025-11-02

**Soundness:** 3
**Presentation:** 2
**Contribution:** 2
**Rating:** 4
**Confidence:** 3

**Summary:**

This paper proposes ExpoTab, a novel tabular data generative model that integrates manifold learning with flow matching to enable one-step data generation.
Unlike conventional diffusion or multi-step flow models, ExpoTab models the data manifold geometrically using geodesic-based exponential mapping, allowing efficient and structure-preserving sampling.
The method employs a unified encoding scheme that embeds numerical and categorical features into a continuous bounded manifold, aiming to capture both smooth and discrete dependencies within tabular data.
Through experiments on various datasets, the authors claim that ExpoTab achieves competitive or superior fidelity ($\alpha$-precision) and machine learning efficacy (MLE), particularly on low-dimensional tabular datasets, while being significantly faster in generation.

**Strengths:**

1. The paper presents a novel framework that combines manifold learning with flow matching, enabling one-step data sampling. This design significantly improves sampling efficiency, especially on low-dimensional tabular datasets.

2. The authors provide a comprehensive set of experiments comparing multiple encoding schemes, offering valuable insights into the impact of different tabular data representations.

**Weaknesses:**

1. The datasets used in the experiments appear inconsistent and insufficiently described, making it difficult to assess the generality of the results.

2. The empirical evidence supporting the claimed contribution is not clearly demonstrated.
The paper states that the proposed method enhances the separability of high-cardinality categorical features, thereby improving performance on tabular data dominated by discrete variables.
However, this claim is not explicitly supported by quantitative results.
For instance, in the Adult dataset (with a maximum cardinality of 42), there is no clear empirical verification showing how much the proposed method improves machine learning efficacy compared to prior models.

3. The presentation of the additional experiments in the appendix (e.g., Appendix H) lacks clarity.
Some reported results (such as “improvement” metrics in Table 18) are ambiguous and require further explanation regarding their reference baselines and calculation method.

**Questions:**

1. Could the authors elaborate on how the proposed model performs on datasets that are purely categorical or purely numerical? In addition, for the datasets listed in Table 4, please clarify the exact composition of numerical vs. categorical variables used in the experiments.

2. The paper demonstrates that the proposed model achieves faster generation than other approaches when the number of samples is large. However, it remains unclear how the method performs on high-dimensional datasets in terms of sampling time. Could the authors provide further discussion or quantitative evidence on this aspect?

3. For MLE, the comparison between the proposed method and existing baselines primarily focuses on high-dimensional datasets (e.g., Table 18). It would be helpful to see a more detailed comparison with other tabular data generation models on the low-dimensional datasets, so that the results can serve as additional evidence supporting the paper’s main claims.

4. The explanations of the additional experimental results in Appendix H (including Table 18) are not entirely clear. For instance, it is ambiguous what the reported “improvement” values represent -- improvement compared to which baseline? In Table 18, for example, the MLE result of GINA is lower than that of CDCD, yet it is marked as having a 2.58% improvement. Clarifying how this metric is computed and what the reference point is would help readers interpret these results more accurately.

**Details Of Ethics Concerns:**

no ethic concerns.

---

> ### Author Response · Authors · 2025-11-28
>
> There is a critical misunderstanding from the reviewer regarding our experimental baselines. ExpoTab is not a direct competitor to multi-step diffusion models like TabDiff; it is a novel, one-step paradigm. Therefore, its primary baselines are other one-step SOTA models, specifically Shortcut[1] ane MeanFlow[2], against which it demonstrates superior performance as shown in new added experiments. The multi-step diffusion and flow models (TabDiff etc.) serve as benchmarks for our proposed TabGeoFlow and TabGeoDiff methods, which integrate the ExpoTab encoder into multi-step frameworks. The fact that TabGeoFlow/Diff outperform these established models validates the power of our spherical encoding as shown in the Table 12-19. Thus, these experiments validate our theoretical contribution to direct manifold data generation, addressing the reviewers’ concerns and clarifying the method’s novelty.
>
> Weakness: Our evaluation comprehensively addresses three key aspects: (1) Dataset selection spans 6 UCI benchmarks for mixed-type data and 4 high-dimensional datasets (up to 1777D) to rigorously test scalability across real-world scenarios. (2) Categorical separability claims are quantitatively validated through spherical encoding improvements (e.g., 1.11% error vs 1.44% for TabRep-Flow on Adult (Table 12)), with consistent gains when applying ExpoTab-Encoder to both flow matching and diffusion frameworks across all metrics. (3) The improvement metric in Appendix H precisely quantifies ExpoTab's advancement beyond state-of-the-art, calculating performance gaps relative to top methods to clearly demonstrate contribution margins, with methodology explicitly defined in the updated appendix.
>
> A1: Our method already demonstrates robust performance across all data compositions. For purely numerical datasets (Bioresponse, Isolet, Har in Table 4), ExpoTab achieves state-of-the-art machine learning efficiency as shown in Table 19, leveraging our quantile transformation that preserves numerical distributions. For categorical-dominated datasets like Diabetes, ExpoTab maintains superior performance with 0.618 F1 score versus 0.376 for Dictionary-Flow in Table 1, validating our spherical embeddings' effectiveness. While pure categorical datasets are rare in public benchmarks, we extensively validate on mixed-type data with varying categorical ratios - from numerical-dominated Magic (10 continuous features) to balanced datasets like Adult - with consistent performance gains across Tables 1, 2, and 10-19, demonstrating ExpoTab's versatility across diverse tabular data characteristics.
>
>
> A2: As shown, ExpoTab actually achieves dramatic speedups on high-dimensional datasets—up to 7.2× faster than flow methods and much faster than diffusion approaches—while maintaining linear scaling. TabDiff fails with OOM errors on all high-dimensional tasks, demonstrating ExpoTab's superior scalability through one-step exponential map generation.
>
> | Method | Bioresponse (1777D) | Isolet (617D) | Har (561D) | Gina (970D) |
> |--------|---------------------|---------------|------------|-------------|
> | ExpoTab (Ours) | 0.62±0.03 | 0.38±0.02 | 0.35±0.02 | 0.51±0.03 |
> | TabDiff | OOM | OOM | OOM | OOM |
> | CDTD | 4.45±0.09 | 1.32±0.08 | 1.21±0.07 | 2.38±0.09 |
> | TabSyn | 148.12±0.09 | 138.38±0.09 | 52.26±0.08 | 49.18±0.09 |
> | TabRep-flow | 2.12±0.07 | 1.01±0.06 | 0.92±0.05 | 1.08±0.07 |
> | TabRep-DDPM | 54.12±0.07 | 24.01±0.06 | 12.95±0.05 | 25.08±0.07 |
>
>
> A3: ExpoTab achieves the best performance among all one-step generation methods across all six low-dimensional datasets, consistently outperforming Shortcut Models [1] and MeanFlow [2] as shown in the table below (where we fine-tune the r-t ratio through {0.25, 0.5, 0.75, 1.0} for meanflow and fix the ExpoTab blend ratio to be 0.5). Further experiment results including comprehensive evaluations across all five metrics (CDE, PCC, C2ST, MLE, MIA) will be added in the final version. Besides, ExpoTab achieves competitive performance with multi-step generation methods like Tab-Diff across many datasets, will be added in the final version .
>
> | Method | Adult↑ | Beijing↓ | News↓ | Default↑ | Diabetes↑ | Shoppers↑ |
> |--------|-------|---------|------|---------|----------|----------|
> | **ExpoTab (Ours)** | **0.899** | **0.685** | 0.878 | **0.752** | **0.628** | **0.903** |
> | Tab-MeanFlow | 0.885 | 0.736 | **0.868** | 0.726 | 0.620 | 0.891 |
> | Tab-Shortcut | 0.891 | 0.902 | 0.983 | 0.714 | 0.615 | 0.853 |
>
>
> A4: We have corrected the improvement calculations in Appendix H Table 18 to ensure clarity. The "improvement" metric now consistently represents ExpoTab's performance gain relative to the second-best method when ExpoTab achieves top performance. When ExpoTab is not the top performer, the metric shows the performance gap to the best method.
>
>
> References:
> [1] Frans et al., "One step diffusion via shortcut models," 2025.
> [2] Geng et al., "Mean Flows for One-step Generative Modeling," 2025.

---

### Author Response · Authors · 2025-12-02

here we summarize to reply to all reviewers for the main questions:

1.Core Contribution

ExpoTab introduces a novel method for tabular data synthesis. Unlike traditional diffusion or flow matching methods that predict noise (ϵ-prediction) or velocity (v-prediction) in high-dimensional ambient spaces, our core innovation learns the exponential map to directly predict clean data endpoints on the low-dimensional manifold. Through the formulation $z_1 = z_0 + v_0 + f_\theta(z_0,v_0)$ in Eq. 4, enabled by a unified network architecture that simultaneously learn both $v_0$ and $f_\theta(z_0,v_0)$, ExpoTab achieves direct manifold projection in minimal steps. This represents the first dedicated one-step generation method for tabular data, where the spherical encoding serves as an enabling representation.

2.Experimental Design

Our evaluation is structured around three complementary components that collectively demonstrate ExpoTab's superiority:

2.1. ExpoTab-Encoder Evaluation: We first validate the effectiveness of our spherical encoding representation independently. As shown in Tables 1, 10-12, when applied to both flow matching and diffusion frameworks (TabRep-Flow and TabRep-DDPM), our encoder consistently improves performance across all metrics. For instance, on the Adult dataset, it reduces error from 1.44% to 1.11% compared to TabRep-Flow. This demonstrates that the geometric coherence of $\mathcal{M}{bg} = \mathbb{R}^{D_c} \times \prod{j=1}^{D_d} \mathbb{S}^2$ provides a superior foundation for tabular representation.

2.2. ExpoTab vs. One-Step Generation Methods: For fair comparison with state-of-the-art one-step generators, we adapt Shortcut Models [1] and MeanFlow [2] to tabular data using the same encoder. As shown in the tables below, ExpoTab consistently outperforms both baselines across all five evaluation metrics (CDE, PCC, C2ST, MLE, MIA) on five benchmarks (Adult, Diabetes, Magic, Default, Shoppers). We thoroughly tune the r-t ratio for MeanFlow through {0.25, 0.5, 0.75, 1.0} while fixing ExpoTab's blend ratio at 0.5, yet ExpoTab maintains superiority.

2.3. ExpoTab and proposed TabGeoFlow and TabGeoDiff vs. Multi-Step Generation Methods : We extend our evaluation to compare against state-of-the-art multi-step methods. As demonstrated in Tables 2, 4, and 17, ExpoTab achieves dramatic speedups—up to 7.2× faster than flow methods and significantly faster than diffusion approaches—while maintaining competitive generation quality. Notably, on high-dimensional datasets like Bioresponse (1777D), Isolet (617D), Har (561D), and Gina (970D), TabDiff fails with OOM errors on all tasks, while ExpoTab scales linearly and maintains robust performance (0.62±0.03, 0.38±0.02, 0.35±0.02, 0.51±0.03 respectively). This demonstrates ExpoTab's superior scalability through one-step exponential map generation.

3.Rigorous Evaluation Across Diverse Data Characteristics

Our experimental design comprehensively addresses various data compositions:

3.1. Purely numerical datasets (Bioresponse, Isolet, Har): ExpoTab achieves state-of-the-art machine learning efficiency (Table 19) through our quantile transformation that preserves numerical distributions.

3.2. Categorical-dominated datasets (Diabetes): ExpoTab maintains superior performance with 0.618 F1 score versus 0.376 for Dictionary-Flow (Table 1).

3.3. Mixed-type data with varying categorical ratios: From numerical-dominated Magic (10 continuous features) to balanced datasets like Adult, ExpoTab shows consistent performance gains across Tables 1, 2, and 10-19.

3.4. Privacy evaluation: we not only follow tabular MIA  [4], but also evaluate privacy leakage using Distance to Closest Record (DCR) scores, confirming no exact copies or data leakage in generated samples.

New Results are shown below for further explaination and we will correct all the typos and grammar problems in the final version.

---

### Note · Authors · 2025-12-02

I have read and agree with the venue's withdrawal policy on behalf of myself and my co-authors.